# NGR-Based Radiopharmaceuticals for Angiogenesis Imaging: A Preclinical Review

**DOI:** 10.3390/ijms241612675

**Published:** 2023-08-11

**Authors:** György Trencsényi, Kata Nóra Enyedi, Gábor Mező, Gábor Halmos, Zita Képes

**Affiliations:** 1Division of Nuclear Medicine and Translational Imaging, Department of Medical Imaging, Faculty of Medicine, University of Debrecen, Nagyerdei St. 98, H-4032 Debrecen, Hungary; trencsenyi.gyorgy@med.unideb.hu; 2ELKH-ELTE Research Group of Peptide Chemistry, Pázmány Péter Sétány 1/A, H-1117 Budapest, Hungary; kata.nora.enyedi@ttk.elte.hu (K.N.E.); gabor.mezo@ttk.elte.hu (G.M.); 3Institute of Chemistry, Faculty of Science, Eötvös Loránd University, Pázmány Péter Sétány 1/A, H-1117 Budapest, Hungary; 4Department of Biopharmacy, Faculty of Pharmacy, University of Debrecen, Nagyerdei St. 98, H-4032 Debrecen, Hungary; halmos.gabor@pharm.unideb.hu

**Keywords:** aminopeptidase N (APN/CD13), angiogenesis, NGR (asparagine-glycine-arginine), preclinical, radioisotope

## Abstract

Angiogenesis plays a crucial role in tumour progression and metastatic spread; therefore, the development of specific vectors targeting angiogenesis has attracted the attention of several researchers. Since angiogenesis-associated aminopeptidase N (APN/CD13) is highly expressed on the surface of activated endothelial cells of new blood vessels and a wide range of tumour cells, it holds great promise for imaging and therapy in the field of cancer medicine. The selective binding capability of asparagine-glycine-arginine (NGR) motif containing molecules to APN/CD13 makes radiolabelled NGR peptides promising radiopharmaceuticals for the non-invasive, real-time imaging of APN/CD13 overexpressing malignancies at the molecular level. Preclinical small animal model systems are major keystones for the evaluation of the in vivo imaging behaviour of radiolabelled NGR derivatives. Based on existing literature data, several positron emission tomography (PET) and single-photon emission computed tomography (SPECT) radioisotopes have been applied so far for the labelling of tumour vasculature homing NGR sequences such as Gallium-68 (^68^Ga), Copper-64 (^64^Cu), Technetium-99m (^99m^Tc), Lutetium-177 (^177^Lu), Rhenium-188 (^188^Re), or Bismuth-213 (^213^Bi). Herein, a comprehensive overview is provided of the recent preclinical experiences with radiolabelled imaging probes targeting angiogenesis.

## 1. Introduction

Aminopeptidase N (APN/CD13) is one of the several angiogenic biomarkers that has recently been exhaustively investigated for both diagnostic and therapeutic purposes [1]. APN/CD13 is a Zn^2+^-dependent transmembrane protease that is responsible for the cleavage of N-terminal neutral amino acids from small peptides [2,3]. It participates in cell adhesion, migration, proliferation, differentiation, invasion, and angiogenic processes [4,5]. Moreover, given its extracellular matrix degradation property, APN/CD13 has a central role in metastasis formation and metastatic spread as well [6].

APN/CD13 is physiologically expressed on the surface of various cells, including macrophages, stromal cells, smooth muscle cells, fibroblasts, and osteoclasts [7,8]. Its expression has also been detected in a wide array of healthy tissues, for example, in the epithelial layer of the intestines, the proximal renal tubules, and the canaliculi of the bile ducts [8]. In addition, APN/CD13 is overexpressed on the activated endothelial cells of blood vessels undergoing angiogenic processes, for example, in tumour-related blood vessels [9]. A high level of receptor expression was found in a broad range of cancers, such as breast, prostate, ovarian, thyroid, pancreatic, colorectal, non-small cell lung cancer, and malignant pleural mesothelioma [7,10,11,12,13,14,15,16,17,18,19]. Although prior literature data state that APN/CD13 cannot be encountered on the surface of normal vasculature [9,20], Curnis et al. reported the existence of different isoforms of CD13 in healthy epithelial cells as well as normal blood vessel endothelium and cancer-linked newly formed vasculature [21].

Phage display technology proved that NGR sequence (Asparagine-Glycine-Arginine) containing peptides show high affinity to APN/CD13 [22]. Additionally, Curnis et al. stated that the APN/CD13 receptors of the healthy blood vessels present different NGR binding capabilities than those of the tumourous neovasculature [21]. Moreover, the NGR motif was reported to exert selectivity for APN/CD13 found on cancer-associated endothelial cells rather than on normal ones [21].

Ample preclinical and clinical evidence strengthens the feasibility of tumour-homing NGR ligands in the identification of receptor positive neoplasms as well as in the imaging of tumour-associated angiogenesis [23,24,25]. Given the easy and cost-effective production and the auspicious imaging characteristics of peptide-based radiotracers, including rapid elimination, fast penetration into tissues, and negligible antigenicity, NGR-based radiopharmaceuticals are considered groundbreaking in the non-invasive, early identification of APN/CD13 positive tumours [22,26,27,28,29]. According to prior studies, several positron emission tomography (PET) and single-photon emission computed tomography (SPECT) radioisotopes could be applied for the labelling of NGR compounds, including Gallium-68 (^68^Ga), Copper-64 (^64^Cu), Technetium-99m (^99m^Tc), Lutetium-177 (^177^Lu), Rhenium-188 (^188^Re), or Bismuth-213 (^213^Bi) (as seen in Figure 1) [30,31,32,33,34,35].

In this review, we provide a comprehensive overview of the in vivo imaging behaviour of the currently existing radiolabelled NGR derivatives specifically designed for the imaging of solid tumour angiogenesis. Moreover, a brief insight is given into the latest advances in the therapeutic application of NGR-based molecules.

## 2. Overview of Preclinical Studies with Radiolabelled NGR-Based Imaging Probes

### 2.1. Labelling with Gallium-68 (^68^Ga)

Out of the positron emitter radiometals, ^68^Ga is the most frequently used isotope for the labelling of NGR probes. Table 1 summarises the preclinical studies with [^68^Ga]Ga-labelled PET radiotracers that selectively target APN/CD13. 

#### 2.1.1. Tracer Uptake in Primary Tumours—First Results

The in vitro stability, lipophilicity, receptor binding capacity, cellular uptake, and in vivo imaging characteristics of NOTA-conjugated G_3_-NGR (Gly_3_-CNGRC; NOTA: 1,4,7-triazacyclononane-triacetic acid) peptide radiolabelled with ^68^Ga ([^68^Ga]Ga-NOTA-G_3_-NGR) were investigated by Shao et al. (see Table 1) [36]. The hydrophilic nature of this probe projects a urinary route of elimination. Although, beyond hydrophilicity, the charge of the peptide groups also determines renal uptake. Further, the attachment of peptides to the luminal part of the renal tubular cells occurs via positively charged groups [48]. Given the positive charge of [^68^Ga]Ga-NOTA-G_3_-NGR_2_—determined by a strong cation exchange cartridge—the renal route of elimination of [^68^Ga]Ga-NOTA-G_3_-NGR could also be explained by the positivity of the molecule [37]. In vitro cellular uptake studies with APN/CD13 expressing HT1080 human fibrosarcoma cells and receptor negative HT29 colon adenocarcinoma cells proved the receptor binding specificity of [^68^Ga]Ga-NOTA-G_3_-NGR. Prompt and selective tumour accumulation of [^68^Ga]Ga-NOTA-G_3_-NGR along with appropriate tumour-to-background ratios—experienced during the static microPET image assessment of female nude BALB/c mice bearing subcutaneous HT1080 tumours in their right upper flank—reinforces the suitability of this tracer in APN/CD13 positive tumour diagnostics. In contrast to the notable radioactivity of the HT1080 tumours, negligible tracer concentration was found in the receptor naive HT29 colon tumour-bearing mice, which was in line with the ex vivo data (4.96 ± 3.18% ID/g and 0.88 ± 0.68% ID/g in HT1080 and HT29 tumourous animals, respectively). Comparable biodistribution patterns were observed in another study by Shao and co-workers, where the in vivo and in vitro comparative characterisation of NGR ([^68^Ga]Ga-NOTA-G_3_-NGR_2_) and RGD-based ([^68^Ga]Ga-NOTA-G_3_-RGD_2_) radiopeptides was carried out in HT1080 tumour xenografts (as presented in Table 1) [37]. The results derived from these preliminary studies opened avenues for further investigation as well as the development of ^68^Ga-labelled NGR probes.

#### 2.1.2. Tracer Uptake in Primary Tumours and Related Metastases

The in vivo tumour diagnostic efficacy of another [^68^Ga]Ga-labelled NGR peptide derivative termed “cyclic NGR” (c(NGR)) was investigated by Máté and co-workers [33]. Their research was centred around the identification of APN/CD13 positivity in primary as well as metastatic tumours using [^68^Ga]Ga-NOTA-c(NGR) (NOTA-c(NGR)/c[Lys(NOTA)-Asn-Gly-Arg-Glu]-NH_2_) in experimental rat model systems. The synthesis procedure of the newly constructed [^68^Ga]Ga-labelled c[KNGRE]-NH_2_ cyclic peptide linked to the chelator *p*-SCN-Bn-NOTA ([^68^Ga]Ga-NOTA-c(NGR)) was also detailed. Cyclic peptides are reported to be more resistant to proteolytic activity (exo- and endoproteases) as well as chemical degradation compared to linear ones [49,50]. The c[KNGRE]-NH_2_ cyclic peptide was the basis of the assessed radiotracer, as it has better chemical stability against deamidation than other existing c(NGR) agents that possess disulfide bridges [51]. Albeit using a disulfide bridge for NGR cyclization is reported to be easier than amide coupling, in research with thermosensitive liposomes linked to c(NGR), Negussie et al. concluded that biodegradation leads to the instability of the disulfide bond of the peptide [37,51]. Furthermore, by performing cyclization via the linkage of an amino group to a carboxyl one, better tumour binding capability could be achieved [51]. Conversely, disulfide bond-based cyclization resulted in the outstanding APN-targeting affinity of c(NGR) peptide (cCPNGRC) with a 30 times lower IC50 (half-maximal inhibitory concentration) for the inhibition of the proteolytic activity of APN/CD13 [52,53].

The in vitro experiments revealed the hydrophilic nature of [^68^Ga]Ga-NOTA-c(NGR) [33]. Confirmed by *LogP* measurements, Dénes et al., Kis et al. and Shao et al. also published the hydrophilic character of different [^68^Ga]Ga-labelled APN/CD13 targeting vectors: [^68^Ga]Ga-NODAGA-c(NGR), [^68^Ga]Ga-NODAGA-c(NGR) (MG1), [^68^Ga]Ga-NODAGA-c(NGR) (MG2), [^68^Ga]Ga-NODAGA-YEVGHRC, [^68^Ga]Ga-NOTA-G_3_-NGR, and [^68^Ga]Ga-NOTA-G_3_-NGR_2_ (as demonstrated in Figure 2) [36,37,38,39,40]. NODAGA-c(NGR), NODAGA-c(NGR) (MG1), and NODAGA-c(NGR) (MG2) refer to c[Lys(NODAGA)-Asn-Gly-Arg-Glu]-NH_2_, c[CH_2_-CO-Lys(NODAGA)-Asn-Gly-Arg-Cys]-NH_2_ and c[CH_2_-CO-Lys(NODAGA)-Asn-N(Me)Gly-Arg-Cys]-NH_2_; respectively (NODAGA: 1,4,7-triazacyclononane,1-glutaric acid-4,7-acetic acid). The *LogP* values of the listed radiopharmaceuticals are displayed in Table 2. In addition, the radio-HPLC (high-performance liquid chromatography) measurements strengthened the stability of the NGR compound in rat serum at 37 °C for 2 h and in PBS (phosphate buffered saline) at 95 °C for 1 h.

Adult male Fischer-344 rats bearing APN/CD13 positive orthotopic and heterotopic transplanted mesoblastic nephroma (Ne/De) tumours were used for the accomplishment of in vivo miniPET and ex vivo biodistribution studies with [^68^Ga]Ga-NOTA-c(NGR). Further, the imaging properties of the NGR derivative were compared with those of the RGD (arginine-glycine-aspartate) ligand containing radiotracer ([^68^Ga]Ga-NODAGA-c(RGD)]_2_) targeting α*_ν_*β_3_ integrin, which is another pivotal biomarker of angiogenesis. The in vivo imaging behaviour and the ex vivo organ distribution were also studied in healthy control Fischer-344 rats. For the heterotopic experimental model, the tumour cells were subcutaneously inoculated into the left thighs of the rats, while for the orthotopic tumour generation, SRCA-based (SRCA: subrenal capsule assay) tumour cell transplantation was performed [54]. The APN/CD13 expression of the tumour cells was verified by Western blot analysis. For the investigation of the in vivo imaging characteristics of the radiotracers, 10-min long whole-body PET scans were performed following the intravenous injection of 7.4 ± 0.2 MBq [^68^Ga]Ga-NOTA-c(NGR) or [^68^Ga]Ga-NODAGA-[c(RGD)]_2_ into the lateral tail vein of both normal control and tumour-bearing rats. Ex vivo data were displayed from the radioactivity measurements of cardiac blood samples and tissue extracts from the liver, spleen, kidneys, intestine, heart, stomach, muscle, lung, and tumour mass. To further verify the APN/CD13 affinity of [^68^Ga]Ga-NOTA-c(NGR), a non-radioactive blocking peptide compound (200 µg of NOTA-c(NGR)) was employed before the application of the radiolabelled match. Thereafter, in vivo and ex vivo uptake inhibition studies were conducted.

In the control animals, discrete [^68^Ga]Ga-NOTA-c(NGR) accumulation was recorded in the investigated abdominal organs, including the liver, the spleen, and the intestines, whereas the kidneys and the urine were depicted with prominent radiotracer accretion. In accordance with the results of Máté et al., Zhang et al.—*investigating the biodistribution pattern of [^68^Ga]Ga-DOTA-NGR*—also experienced slight tracer accumulation, particularly in the abdominal organs (DOTA: 1,4,7,10-teraazacyclododecane-N,N′,N″,N‴-teraacetic acid) [41]. Furthermore, this was in correlation with former biodistribution studies with [^68^Ga]Ga-NOTA-G_3_-NGR and [^68^Ga]Ga-NOTA-G_3_-NGR_2_ conducted by Shao et al. [36,37]. These abdominal accumulation values were lower compared to those of the RDG compound [33]. Although Shao et al. also detected minimal abdominal [^68^Ga]Ga-NOTA-G_3_-NGR_2_ uptake,—unlike the results of Máté et al.—no statistically significant difference could be noted between the in vivo organ distribution characteristics of the NGR and the RGD derivatives [37]. Both the in vivo and ex vivo figures confirm that the radiopharmaceutical is excreted via the urinary system, which is further supported by the *LogP* value. Similarly, previous research on different [^68^Ga]Ga-labelled c(NGR) peptides such as [^68^Ga]Ga-NODAGA-c(NGR), [^68^Ga]Ga-NODAGA-c(NGR) (MG1), [^68^Ga]Ga-NODAGA-c(NGR) (MG2), and [^68^Ga]Ga-NODAGA-YEVGHRC also found renal ways of clearance [38,39,40]. Supressing studies with non-radioactive NOTA-c(NGR) showed two times lower tracer uptake upon the administration of the blocking agent, suggesting the APN/CD13 target selectivity of the probe.

Assessing the miniPET scans of the Ne/De tumourous rats, the subcutaneously growing tumours could definitely be detected with the investigated NGR compound. The high tumour-to-muscle (T/M) ratios made the exact demarcation of the tumour tissue from the background possible (T/M mean standardised uptake value (SUV_mean_): 12.25 ± 3.08 and T/M SUV_max_: 16.28 ± 2.48). In this study, two times lower uptake values were found for the RGD peptide (T/M SUV_mean_: 6.39 ± 0.38 and T/M SUV_max_: 7.8 ± 1.14) than in case of the NGR compound; however, in the experiment of Shao et al., the RGD and NGR tumour uptake of the subcutaneous HT1080 xenografts did not differ significantly (6.42 ± 2.21, 5.43 ± 2.76, and 4.02 ± 2.03% ID/g for [^68^Ga]Ga-NOTA-G_3_-NGR_2_ and 7.84 ± 1.94, 6.26 ± 1.63, and 5.13 ± 1.88% ID/g for [^68^Ga]Ga-NOTA-G_3_-RGD_2_; respectively 0.5, 1 and 2 h post-injection) [37]. 

The tumour targeting potential of [^68^Ga]Ga-NOTA-c(NGR) was further verified by in vivo blocking experiments that showed that the quantitative uptake figures of the tumours significantly diminished after the application of the blocking molecule (T/M SUV_mean_ blocked: 2.50 ± 0.39 vs. unblocked: 12.25 ± 3.08 and T/M SUV_max_ blocked: 2.48 ± 1.14 vs. unblocked 16.28 ± 2.48).

The diagnostic ability and the APN/CD13 binding selectivity of [^68^Ga]Ga-NOTA-c(NGR) were also authenticated in the SRCA tumour model by the in vivo PET distribution examinations and the in vivo blocking studies, respectively. Consistent with the findings of Máté et al., the diagnostic feasibility of [^68^Ga]Ga-NOTA-c(NGR) was also confirmed by Szabó and colleagues in SRCA-induced Ne/De tumours [42]. Comparing the uptake of the NGR and RGD molecules, considerable disparity was registered between the SUV_max_ and the T/M SUV_max_ values of [^68^Ga]Ga-NOTA-c(NGR) (SUV_max_ and T/M SUV_max_: 11.47 ± 1.37 and 12.47 ± 2.19; respectively) and [^68^Ga]Ga-NODAGA-[c(RGD)]_2_ (SUV_max_ and T/M SUV_max_: 4.96 ± 0.47 and 6.98 ± 0.55; respectively). On the contrary, no remarkable difference was shown between the T/M ratios of the NOTA-linked [^68^Ga]Ga-labelled dimeric Gly_3_-CNGRC cyclic peptide ([^68^Ga]Ga-NOTA-G_3_-NGR_2_) and [^68^Ga]Ga-NOTA-G_3_-RGD_2_ in the study of Shao et al. applying female BALB/c mice bearing HT1080 tumours with simultaneous expression of APN/CD13 and α_ν_β_3_/α_ν_β_5_ integrin [37,55,56].

Identically to the observations experienced in the case of the subcutaneously growing primary Ne/De tumours, the SUV values of the subrenally located tumours notably decreased (*p* ≤ 0.01) after the injection of the unlabelled compound (T/M SUV_mean_ blocked: 1.71 ± 0.24 vs. unblocked: approx. 9 and T/M SUV_max_ blocked: 1.96 ± 0.56 vs. unblocked: 12.47 ± 2.19) that authenticates the APN/CD13 adhering capacity of the tracer. These blocking results—similarly to prior literature findings on other [^68^Ga]Ga-labelled NGR-based molecular probes—suggest that these diagnostic agents specifically target the APN/CD13 receptors [36,37,38,41].

Of note, the subrenally transplanted primary Ne/De tumours exhibited increased radioactivity in comparison with the subcutaneous tumours. The suboptimal heterotopic tumour niche lacking in appropriate vasculature for tumour cell growth and metastatic spread may lead to the difference observed between the tracer accumulation of the two primary tumour models [57,58].

To test the efficacy of the NGR-based peptides in metastasis identification, Máté et al. performed ex vivo organ distribution studies as well as blocking experiments in the APN/CD13 positive mesenteric lymph node and parathymic lymph node metastases of the primary Ne/De tumours. The ex vivo DAR (differential absorption ratio) values demonstrated notable [^68^Ga]Ga-NOTA-c(NGR) uptake in both types of lymph nodes (mesenteric and parathymic lymph nodes: 0.30 ± 0.18 and 0.69 ± 0.24, respectively). Correspondingly, the parathymic lymph node metastases derived from the SRCA-generated Ne/De tumours were also clearly visible with [^68^Ga]Ga-NOTA-c(NGR) in the recent experiment of Szabó et al. [42].

Similarly to the previous findings, the successful prohibition of tracer uptake proved the APN/CD13 targeting specificity of the [^68^Ga]Ga-labelled tracer.

Both the primary and metastatic tumours exhibited considerable NGR tracer uptake, and more enhanced [^68^Ga]Ga-NOTA-c(NGR) accumulation was noted in the Ne/De tumours in comparison with the uptake of [^68^Ga]Ga-NODAGA-c(RGD)]_2_. Given the difference between the T/M values of the RGD and NGR-based probes, we may state that the angiogenic targeting ability of [^68^Ga]Ga-NOTA-c(NGR) surpasses that of the RGD compound. The above-detailed results of Máté and colleagues further strengthen the idea that the [^68^Ga]Ga-labelled NGR-motif containing diagnostic vectors could be potential weapons not only in the timely identification of primary tumours with high APN/CD13 density but also in related metastatic diseases. Moreover, the negligible background activity along with the prominent uptake of the tumours ensure the acquisition of high-contrast images, which is of critical importance regarding PET scan reporting. An overview of the study by Máté et al. is given in Table 1.

#### 2.1.3. Temporal Evolution of Radiopharmaceutical Uptake with Tumour Growth

Also using [^68^Ga]Ga-NOTA-c(NGR), Kis and co-workers dealt with the in vivo assessment of the temporal changes of the APN/CD13 pattern in rat hepatocellular carcinoma (He/De) [39]. In addition, the comparison of APN/CD13-specific [^68^Ga]Ga-NOTA-c(NGR) with α_ν_β_3_ integrin-affine ^68^Ga-NODAGA-[c(RGD)]_2_ was also carried out. For the accomplishment of their study, rat hepatocellular (He/De) carcinoma models [59] were established by the subcutaneous injection of 5 × 10^6^ He/De cells into the left scapular region of Fischer-344 rats. Similarly to the findings of Máté et al. with Ne/De tumours, PET data showed that the [^68^Ga]Ga-NOTA-c(NGR) uptake of the primary experimental hepatic (He/De) tumours was more elevated relative to the accumulation of [^68^Ga]Ga-NODAGA-[c(RGD)]_2_ [33]. Strengthened by western blot analysis, this finding could be attributed to the higher density of the APN/CD13 molecules in He/De tumours [39]. Unlike previous literature data, however, remarkably heterogenous radiopharmaceutical accumulation was visualised in the subcutaneously transplanted He/De tumours using both angiogenesis-directed radioprobes. The expression of angiogenic biomarkers within a tumour mass is dependent upon the severity of hypoxia, the dominance of tumour-related inflammation, and genetic instability [60]. Thus, tumour heterogeneity and associated heterogenous vasculogenesis result in heterogenous uptake patterns. Given the increasing uptake of [^68^Ga]Ga-NOTA-c(NGR) in relation to tumour growth, Kis et al. confirmed the relationship between the size extension of angiogenic tumour regions and the enlargement of the whole tumour mass. Furthermore, based on the meaningful connection between the accumulation of hypoxia-specific [^68^Ga]Ga-DOTA-nitroimidazol and [^68^Ga]Ga-NOTA-c(NGR), *Kis and colleagues* stated that the extent of hypoxia is proportional to tumour progression and angiogenic processes. Contrarily—evaluating the expression profile of α_v_β_3_ integrin, endoglin, and vascular endothelial growth factor receptor 2 (VEGFR2) in subcutaneous breast, ovarian, and pancreatic cancer xenografts—Deshpande et al. published a reverse correlation between tumour size and the presence of angiogenic biomarkers [61]. The differences between the examined experimental tumour models, the use of pro-angiogenic Matrigel, and the way of tumour generation may underpin the contradiction between the results of Kis et al. and those of Deshpande and co-workers [61]. These research findings—in line with the prior study of Máté et al.—collectively strengthen the suitability of [^68^Ga]Ga-NOTA-c(NGR) as an imaging probe in the diagnostic armamentarium of angiogenesis imaging. Moreover, [^68^Ga]Ga-NOTA-c(NGR)-based tracking of the change in the appearance of APN/CD13 over time may be the basis for efficient anti-tumour therapeutic planning. A summary of this study is presented in Table 1.

#### 2.1.4. Comparison of Various ^68^Ga-labelled NGR-Based Radiotracers

In another positron emission tomography/magnetic resonance imaging (PET/MRI) experiment by Kis et al. the APN/CD13 target selectivity of the following four different [^68^Ga]Ga-labelled NGR peptide compounds was compared using chemically induced syngeneic He/De and Ne/De preclinical rat tumour models: [^68^Ga]Ga-NOTA-c(NGR), [^68^Ga]Ga-NODAGA-c(NGR), [^68^Ga]Ga-NODAGA-c(NGR) (MG1) or [^68^Ga]Ga-NODAGA-c(NGR) (MG2) (as exhibited in Table 1) [38]. NODAGA-c(NGR), NODAGA-c(NGR) (MG1), and NODAGA-c(NGR) (MG2) refer to c[Lys(NODAGA)-Asn-Gly-Arg-Glu]-NH_2_, c[CH_2_-CO-Lys(NODAGA)-Asn-Gly-Arg-Cys]-NH_2_ and c[CH_2_-CO-Lys(NODAGA)-Asn-N(Me)Gly-Arg-Cys]-NH_2_; respectively. For this purpose, in vivo PET/MRI imaging and ex vivo organ distribution assessment took place in 16 weeks old male Fischer rats bearing He/De [59] or Ne/De [33] tumours in their left scapular region. In correlation with the earlier study of Máté et al., the APN/CD13 positivity of the Ne/De and He/De tumours was authenticated by immunohistochemical staining [33,38].

Since the choice of the most appropriate NGR-based radiotracer is crucial not only in diagnostics but also in therapeutic decision making, Kis et al. selected NGR molecules with different chemical and metabolic stability to test the effects of these variabilities on image quality. Additionally, the radiotracer accumulation is not only determined by the chemical structures of the NGR peptides but also by the applied chelators [46,62].

In line with available literature findings on [^68^Ga]Ga-DOTA-NGR, [^68^Ga]Ga-NOTA-NGR, and [^68^Ga]Ga-NOTA-c(NGR), the [^68^Ga]Ga-labelled c(NGR) derivatives showed increased accumulation in the kidneys, whereas discrete radiotracer concentration was found in the abdominal and thoracic organs [33,36,37,39,41]. In a similar manner, the biodistribution studies of Satpati et al. displayed very low abdominal accumulations of [^68^Ga]Ga-NODAGA-NGR [46]. Upon evaluation of the PET/MRI scans, the He/De and Ne/De tumours could be visualised with all investigated NGR-based tracers. This was in correlation with the former study of Máté et al. conducted with primary subcutaneous Ne/De and SRCA-induced Ne/De tumour models [33]. Additionally, other experiments of the same research group also confirmed the diagnostic potential of [^68^Ga]Ga-NOTA-c(NGR) in He/De tumour-bearing rats [39] and [^68^Ga]Ga-NODAGA-c(NGR) in mice bearing HT1080 and B16F10 tumours [63]. However, comparing the tumour accumulation kinetics of the four radioprobes, meaningful differences were encountered. The tumour uptake of the NODAGA-chelated MG1 and MG2 molecules was lower in comparison with that of the other two c(NGR) compounds. Although the exact mechanism behind this is not yet fully covered, the structural differences between the tracers may partially explain these findings. Moreover, the severity of hypoxia, the tumour types, and the continuous change of the receptor profile of the tumours may also have an impact on radiotracer accretion [61].

#### 2.1.5. CendR Sequence-Containing NGR-Based Probe

To assess the suitability of the newly constructed internalizing NGR (iNGR) peptide—*that contains a target specific NGR motif and a tumor-penetrating CendR sequence (C-end rule peptide; R/KXXR/K)*—for the imaging of APN/CD13 overexpressing tumours, Zhao et al. carried out the comparative performance evaluation of [^68^Ga]Ga-labelled iNGR and NGR (demonstrated in Table 1) [43]. 

Possessing a C-terminal arginine or lysine with a free carboxyl group, the CendR motif (R/KXXR/K) enhances the circulation of the peptide and regulates its movement across biological membranes through its engagement with neuropilin-1 (NRP-1) [64,65,66]. The CendR sequence also favours the permeation of various agents, including chemicals and different nanoparticles, into tissues by interacting with NRP-1 [67]. Initiated by the success of internalizing RGD (iRGD)-attached nanoparticles in augmenting the sensitivity of tumour targeting imaging molecules as well as the efficacy of anti-cancer drugs, Alberici et al. applied iNGR-covered peptides to improve the tumour penetration of chemotherapeutic agents [65,68]. 

Zhao et al. applied DOTA for the conjugation of the investigated NGR derivatives. CD13 expressing HT1080 and receptor-negative HT29 mouse xenograft models were established for the in vitro and in vivo imaging characterisation of [^68^Ga]Ga-DOTA-iNGR and [^68^Ga]Ga-DOTA-NGR [43].

The competitive cell binding affinity study revealed no effects of the CendR motif either on the structure or the APN/CD13 selectivity of the NGR peptide. Higher iNGR uptake of the HT1080 cells—experienced at every investigated time point of the cell binding assay where the mechanism of NRP-1 could be expressed—indicates the better tumour-homing ability of the internalising peptide compared to the NGR counterpart.

The HT1080 tumours exhibited higher iNGR uptake in comparison with NGR in the microPET images ([^68^Ga]Ga-iNGR: 3.41 ± 0.28, 2.97 ± 0.30, and 2.64 ± 0.31%ID/g at 0.5, 1, and 1.5 h post-injection; respectively, [^68^Ga]Ga-NGR: 2.68 ± 0.35, 1.91 ± 0.32, and 1.45 ± 0.30 %ID/g at 0.5, 1, and 1.5 h post-injection; respectively). The enhanced in vivo (1.6 times higher 1 h after injection) and ex vivo (1.9 times higher) tumour uptake of the iNGR derivative is consistent with previous research, where the two-fold higher cellular uptake ratio of the dual-modified liposomes was due to attachment to NRP-1 [69]. Likewise, the tumour accumulation of the iNGR-coated phage vector was three times higher than that of the NGR-coupled one [68]. These studies highlight that the mechanism of action of NRP-1 is the same regardless of the size of the investigated molecule, since identically elevated uptake ratios were experienced both in the case of more sizeable agents (for example, liposomes) and smaller radiotracers. Therefore, the CendR motif can be attached to an extended spectrum of molecules, which ensures its wider application and more probable clinical transportation.

Analysing the results of the uptake inhibition studies, comparable iNGR and NGR concentrations were found in the HT1080 tumours post application of a neutralising NRP-1 antibody, which strengthens the role of NRP-1 in the internalization of iNGR. This is in agreement with former research data that explored the central contribution of NPR-1 to the higher retention of iNGR [68]. In correlation with other studies, minimal tracer accumulation was found in the rest of the investigated organs except for the kidneys. Given the increased tumour accretion and the superior tumour retention (1.7 times higher 1.5 h post-injection) of [^68^Ga]Ga-DOTA-iNGR compared to [^68^Ga]Ga-DOTA-NGR through NRP-1, the incorporation of the CendR motif into the structure of the peptide appears to be a valuable means for the enhancement of the imaging characteristics of the NGR-based probes. Beyond the integration of tumour-penetrating sequences (CendR) into the peptide base, dimeric preparation, and alteration of non-binding sites, multi-receptor-directed heterodimerization could also improve both the uptake and the retention of tumour affine vectors [70,71,72,73]. Moreover, the groundbreaking findings of Zaho et al. may open a novel era in the establishment of tumour-directed diagnostic molecules containing peptide sequences with high tissue-penetrating ability. These tumour-specific probes could not only be exploited in imaging settings but would also serve as promising tools for targeted therapeutic applications.

Similarly to Zhao et al., Zhang and co-workers also used DOTA for the chelation of the NGR peptide [41]. Although the radiotracer applied in their study does not contain any additional sequences, given that the same chelator was attached to NGR as in the study of Zhao et al., we thought it noteworthy to provide a brief discussion on their results here. According to the biodistribution results, the uptake pattern of [^68^Ga]Ga-labelled DOTA-NGR was largely comparable to that of [^68^Ga]Ga-NOTA-c(NGR) [41]. Similarly to the findings of Dénes et al., Kis et al., and Máté et al., the biodistribution data revealed that the elimination of [^68^Ga]Ga-DOTA-NGR occurs mostly through the kidneys [33,38,39,40]. Due to the insignificant background accumulation and the notably high in vivo uptake of the APN/CD13 positive A549 tumours, PET images of high standard could be gathered. As mentioned above, previous experiments with various NGR peptide compounds also noted relatively insignificant radioactivity, mainly in the investigated abdominal organs [33,36,37]. Hence, we might draw the conclusion that the replacement of the chelator from NOTA/NODAGA *([^68^Ga]Ga-NOTA-c(NGR)/[^68^Ga]Ga-NODAGA-c(NGR))* to DOTA *([^68^Ga]Ga-DOTA-NGR)* neither has any influence on the tumour-to-background ratios nor on the subsequent PET image quality. The strong association between [^68^Ga]Ga-DOTA-NGR accumulation and the APN/CD13 density of the A549 tumour xenografts—corresponding to other investigations—suggests the binding specificity of [^68^Ga]Ga-DOTA-NGR to APN/CD13 [6,55]. In addition, compared to other NGR-based probes furnished with imaging labels other than ^68^Ga—for example, Copper-64 (^64^Cu) or Technetium-99m (^99m^Tc)—[^68^Ga]Ga-DOTA-NGR presented improved biodistribution and pharmacokinetical profiles [6,55]. Overall, the results of Zhang and colleagues also proved that [^68^Ga]Ga-labelled NGR derivatives serve as promising agents in the detection of APN/CD13 overexpressing tumours [41]. Taking the better organ distribution properties and the stability of [^68^Ga]Ga-DOTA-NGR compared to [^64^Cu]Cu-DOTA-NGR into consideration, the radiolabelling of the NGR motif containing peptides with ^68^Ga might be superior to labelling with ^64^Cu. This research by Zhang and co-workers is detailed in Table 1. Table 3 displays and compares the physical properties of ^68^Ga and ^64^Cu.

#### 2.1.6. Linear NGR Peptide-Based Probe

Taking into account that the NGR sequence is prone to the deamidation of the Asn side chain, the establishment of less vulnerable APN/CD13 specific peptides is of pivotal importance in angiogenesis imaging [74]. Initial research findings are already available on the diagnostic performance of novel linear sequences directed towards angiogenesis associated biomarkers [40,75]. In a former proof-of-concept study by Dénes et al., the in vivo behaviour of [^68^Ga]Ga-labelled c(NGR) peptide and linear LN peptide YEVGHRC (tyrosyl-glutamyl-valyl-glycyl-histidyl-arginyl-cysteine)-based radiotracers was studied at the preclinical level (seen in Figure 2) [40]. Table 1 displays the most important characteristics of their research.

With the aim of exploring the in vivo angiogenesis-targeting capability of the newly constructed APN/CD13 affine [^68^Ga]Ga-NODAGA-YEVGHRC and [^68^Ga]Ga-NOTA-c(NGR), 12 weeks old male C57BL/6 mice bearing B16F10 melanoma tumours in their left shoulder area were applied. The APN/CD13 positivity of the B16F10 tumour cell lines is well-established [45,76,77]. Eight-to-nine days post B16F10 melanoma tumour cell transplantation, both tumourous and tumour naive healthy mice received 5.5 ± 0.7 MBq of [^68^Ga]Ga-NOTA-c(NGR) or [^68^Ga]Ga-NODAGA-YEVGHRC and were subjected to a 20 min long static PET examination. Representative decay-corrected coronal PET images of healthy control mice are displayed in Figure 2. To better assess the biodistribution of both [^68^Ga]Ga-NODAGA-YEVGHRC and [^68^Ga]Ga-NOTA-c(NGR), the radioactivity of the blood and urine samples as well as different organs and tissues of interest was measured. According to the urinary route of excretion, PET data of the control small animals showed increased renal tracer uptake 90 min after the injection of both assessed probes (as presented in Figure 2). In line with this, previous studies of Kis et al. and Máté et al. also reported renal elimination in the case of the following APN/CD13-targeting [^68^Ga]Ga-labelled c(NGR) derivatives: [^68^Ga]Ga-NOTA-c(NGR), [^68^Ga]Ga-NODAGA-c(NGR), [^68^Ga]Ga-NODAGA-c(NGR) (MG1), and [^68^Ga]Ga-NODAGA-c(NGR) (MG2) [33,38,39]. No significant difference was found between the ex vivo uptake values of [^68^Ga]Ga-NOTA-c(NGR) and [^68^Ga]Ga-NODAGA-YEVGHRC in any of the selected organs and tissues (*p* ≤ 0.05). Although the subcutaneously growing tumours could be clearly delineated from the surrounding tissues with both angiogenesis-directed probes, the c(NGR) containing tracer showed higher tumour uptake ([^68^Ga]Ga-NOTA-c(NGR) SUV_mean_: 0.12 ± 0.02) than the YEVGHRC compound ([^68^Ga]Ga-NODAGA-YEVGHRC SUV_mean_: 0.03 ± 0.01). Therefore, based on these uptake values, the tumour specificity of the labelled YEVGHRC derivative was not appropriate. Structural modifications or the addition of targeting sequences to the molecule may improve its imaging properties. Conversely, Jia et al. strengthened the APN/CD13 tumour-homing ability of YEVGHRC peptide-functionalised liposomes [75]. They reported that such liposomes could serve as useful tools to carry nano-encapsulated drugs to APN/CD13 expressing tumours under in vitro and in vivo conditions. In accordance with the in vivo outcomes, the ex vivo data also confirmed higher [^68^Ga]Ga-NOTA-c(NGR) accumulation (%ID/g: 1.60 ± 0.30) in the tumourous alterations relative to [^68^Ga]Ga-NODAGA-YEVGHRC (%ID/g: 0.13 ± 0.06). Studying the target specificity of various [^68^Ga]Ga-labelled NGR-based imaging molecules in hepatocellular carcinoma (He/De), mesoblastic nephroma (Ne/De), and melanoma tumours, Kis et al., Máté et al., and Satpati et al. reported comparable findings to those of Dénes et al. [33,38,45]. Of note, [^68^Ga]Ga-labelled cyclic NGR (cyclic(lysyl-asparaginyl-glycyl-argininyl-glutamic acid amide)/c(KNGRE)-NH_2_) was found to present the foremost tumour-targeting affinity. Although the optimisation of the pharmacokinetics of YEVGHRC tracers is part of future work, the pioneering observations of Dénes et al. may initiate further investigations on more resistant forms of targeting vectors.

#### 2.1.7. Dimeric NGR Peptide-Based Probes

With the aim of improving the tumour diagnostic potential of [^68^Ga]Ga-labelled NGR-based radiopharmaceuticals, Israel et al. intended to evaluate the effect of NGR dimerisation on the imaging characteristics of the radiolabelled APN/CD13 targeting compounds [47]. Pioneering studies with NGR dimers are already available from 2014 [37]. According to literature findings, since dimeric NGR peptides are able to simultaneously bind to multiple receptor sites in the target cells, the dimers are featured with an increased overall APN/CD13 binding affinity than the monomer targeting ligands [6,55]. For this reason, Isreal et al. aimed to compare the target specificity of NGR-based radiopharmaceuticals with one and two NGR sequences ([^68^Ga]Ga-NODAGA-NGR (NGR monomer) vs. [^68^Ga]Ga-NOTA-(NGR)_2_ (NGR dimer)) under in vitro and in vivo circumstances [47]. The most significant points of their work are shown in Table 1.

Fluorescence-activated cell sorting (FACS) was used for the identification of APN/CD13 expression in the following cell lines: A549 (human lung carcinoma), SKHep-1 (human liver adenocarcinoma), and MDA-MB-231 (human breast carcinoma). APN/CD13 positivity was registered in the A549 (64.3% ± 4.7%) and SKHep-1 (99.3% ± 0.6%) cells, while the MDA-MB-231 cells did not show meaningful CD13 expression.

The cellular uptake of the two investigated imaging probes was evaluated in vitro. Following the incubation of each cell line with [^68^Ga]Ga-NODAGA-NGR and [^68^Ga]Ga-NOTA-(NGR)_2_ for 15, 30, or 60 min, a gamma-counter was used for the determination of the cellular radiotracer accumulation. Although increased radiopharmaceutical accretion was recorded in all three cancer cell lines investigated with the NGR dimer in comparison with its monomeric counterpart, both probes demonstrated low in vitro uptake (less than 1%).

Ten-minute-long microPET imaging studies were performed in eight-week-old *CD1-Foxn1nu*-mice bearing A549 or SKHep-1 tumours in their front flank to study the in vivo diagnostic efficacy of [^68^Ga]Ga-NODAGA-NGR and [^68^Ga]Ga-NOTA-(NGR)_2._ Since the best tumour-to-background ratios could be attained 50 min after tracer injection, the microPET examinations took place 50 min post-administration of 7.8 ± 3.5 or 7.4 ± 3.4 MBq [^68^Ga]Ga]NODAGA-NGR or [^68^Ga]Ga-NOTA-(NGR)_2_, respectively, at an average tumour diameter of ≥0.5 cm. In the case of both assessed imaging probes, 10 A549 and 16 SKHep-1 tumourous mice were injected. Tumour-to-liver, tumour-to-lung, and T/M ratios were registered to ensure quantitative parameters for appropriate lesion detection. Although minor differences could be observed between the in vivo characteristics of the monomer and the dimer NGR, these were statistically not significant. Both APN/CD13 expressing tumour types could be definitely identified with both NGR-based tracers. Upon visual assessment, Israel et al. observed that the monomer NGR derivative seemed to accumulate more in the selected tissues and organs compared to its dimer sister. Moderately higher T/M ratios of the A549 (4.2 ± 1.2 and 5.3 ± 2.0 for the monomer and the dimer NGR, respectively) and SKHep-1 (3.6 ± 1.3 and 4.2 ± 1.5 for the monomer and the dimer NGR, respectively) tumours were observed in the case of the NGR dimer relative to the monomeric match; however, this was statistically not meaningful. Regardless of the fact that the monomer probe tended to present more elevated uptake in both tumours compared to the dimer, based on its higher concentration in the muscle tissue, Israel et al. concluded that improved T/M ratios could be attained using the dimer NGR instead of the monomer one. Tumour-to-liver ratios were depicted to be lower for the NGR dimer than for the NGR monomer in both the A549 and the SKHep-1 tumours. Identically to the T/M ratios, an increased tumour-to-lung ratio was registered for [^68^Ga]Ga-NOTA-(NGR)_2_ (5.1 ± 1.7) than for [^68^Ga]Ga-NODAGA-NGR (4.1 ± 1.0). Besides microPET imaging, the in vivo binding affinities of the radiopharmaceuticals were further evaluated with the administration of the APN/CD13 inhibitor bestatin in the A549 lung tumour-bearing mice. The co-injection of bestatin managed to induce tumour tracer uptake inhibition; however, it was not significant.

In a bid to investigate the radiotracer uptake of different organs and tissues, an ex vivo organ distribution pattern was defined. Healthy control mice were intravenously administered with 6.7 ± 1.1 MBq or 7.6 ± 0.7 MBq of [^68^Ga]Ga-NODAGA-NGR or [^68^Ga]Ga-NOTA-(NGR)_2_; respectively. After the tracer injection, blood samples were gathered every 10 min. Sixty minutes post-injection, various organs were harvested, and these organs and the blood samples were measured for radioactivity using a gamma counter. Based on the ex vivo data, both tracers demonstrated prompt, largely identical clearance from the blood stream. The elimination of the imaging probes occurred mostly through the urinary tract. In line with the in vivo results, the dimer [^68^Ga]Ga-NOTA-(NGR)_2_ showed lower tracer uptake in all investigated organs—*apart from the hepatic and the lienal activity*—in comparison with the monomer [^68^Ga]Ga-NODAGA-NGR. In accordance with the diminished tumour-to-liver ratio of the dimer [^68^Ga]Ga-NOTA-(NGR)_2_, the ex vivo data showed higher hepatic tracer accretion in the case of the dimeric compound, which could be explained by its more prominent hepatobiliary excretion or more lengthened retention in the liver tissue compared to the monomeric derivative.

Immunohistochemical staining also confirmed the presence of APN/CD13 both in the tumours and the tumour-related vasculature, which is in line with the findings observed with the FACS technique.

According to the results of the detailed study, the APN/CD13 positive tumour imaging properties of the monomer and dimer NGR tracers were similar [47]. Therefore, regardless of the presence of an additional NGR sequence, both imaging probes appear to be equally well-suited for the detection of APN/CD13 positive malignancies in vivo. We suppose, however, that by incorporating additional NGR sequences into the radiotracer, the benefits of multivalency could be better exploited.

The multivalency effect of c(NGR) peptide was also assessed in the recent study of Yang and co-workers, who intended to appraise the in vitro and in vivo efficacy of [^68^Ga]Ga-labelled dimeric cNGR ([^68^Ga]Ga-DOTA-c(NGR)_2_) using ES2 and SKOV3 ovarian tumour cell lines and corresponding mouse xenografts (seen in Table 1) [44]. According to the 87.2% and 27.6% APN/CD13 expression rates of the ES2 and SKO3 cells, respectively, higher in vitro [^68^Ga]Ga-DOTA-c(NGR)_2_ accumulation was observed in the ES2 cells than in the SKOV3 cells, which confirmed the receptor selectivity and affinity of the probe. Analysing the microPET scans of nude mice carrying ovarian cancer, [^68^Ga]Ga-DOTA-c(NGR)_2_ exhibited significant accumulation in APN/CD13 overexpressing ES2 tumours (0.62 ± 0.09% ID/g and 0.53 ± 0.08% ID/g at 1 h and 1.5 h, respectively). In line with this result, the same research group had similar findings when the receptor targeting ability of [^68^Ga]Ga-DOTA-NGR was evaluated in intensively receptor positive pulmonary malignancies [41]. Furthermore, noticeable [^99m^Tc]Tc-labelled c(NGRyk) peptide ([^99m^Tc]Tc-MAG_3_-PEG_8_-c(NGRyk)) accumulation was found in vivo in human ovarian adenocarcinoma (OVCAR-3)-bearing nude mice by Faintuch et al. [49]. Moreover, Meng et al. reported the feasibility of Cy5.5-labelled, NGR-conjugated iron oxide nanoparticles (Cy5.5-NGR-Fe_3_O_4_ NPs) in the MR/NIRF dual-modal imaging of ovarian tumour xenografts (MR/NIRF: magnetic resonance/near-infrared fluorescence imaging) [78]. Upon MR imaging, they encountered pronounced tumour T2* signal reduction, while enhanced Fe_3_O_4_-Cy5.5-NGR uptake was recorded by near infrared fluorescence imaging in the neoplastic tissue. Consistent with the above remarked conclusions—*based on the results of* Yang et al.—NGR-based radiolabelled molecular agents are well-suited for the non-invasive characterisation of the receptor distribution of APN/CD13 positive tumours leaving out the invasive procedure of histological tissue sampling and related health ramifications [44,79,80].

### 2.2. Labelling with Copper-64 (^64^Cu)

Radiolabelling of NGR peptides with Copper-64 (^64^Cu) can be a promising alternative to [^68^Ga]Ga-labelling. Chen et al. reported the feasibility of [^64^Cu]Cu-labelled monomeric and dimeric NGR compounds in the non-invasive identification of APN/CD13 receptors at the preclinical level (demonstrated in Table 4) [55]. In correlation with the former synthesis of Zhao et al. and Zhang et al., Cheng and co-workers also applied DOTA for the conjugation of the radiometal with the tripeptide sequence [41,43]. Similarly to previous studies, APN/CD13 overexpressing HT1080 and receptor negative HT29 cell lines and corresponding tumour-bearing athymic nude mice were used to assess the in vitro and in vivo performance of monomeric [^64^Cu]Cu-DOTA-NGR_1_ and dimeric [^64^Cu]Cu-DOTA-NGR_2_ [36,43]. In line with the results of the Western blot analysis and the immunofluorescence staining, the in vitro experiments also showed receptor overexpression in the HT1080 cells, while the HT29 colon adenocarcinoma cells were lacking in APN/CD13. Cellular uptake studies displayed 1.4-fold higher uptake of the dimeric derivative (1.42 ± 0.20%) compared to the monomeric match (1.00 ± 0.08%) in the HT1080 cells, which was consistent with the findings of Israel et al. about [^68^Ga]Ga-labelled monomeric and dimeric NGR peptides [47]. 

Identically to the former results of Israel and colleagues with [^68^Ga]Ga-labelled NGR probes, the HT1080 tumours were well delineated from the background with both the monomeric and the dimeric [^64^Cu]Cu-labelled agents [47,55]. While Israel et al. encountered no statistically significant difference between the in vivo diagnostic potential of the monomer and the dimer NGR, Chen et al. noted improved imaging properties and pharmacokinetics in the case of the [^64^Cu]Cu-DOTA-NGR_2_ [47,55]. Upon the assessment of the in vivo static PET scans, enhanced tumour accumulation and retention were observed for the dimeric NGR at all time points, with respective uptake values of 6.53 ± 0.20%, 6.09 ± 0.18%, 5.22 ± 0.17%, and 3.60 ± 0.23% ID/g at 1-, 2-, 4-, and 24-h post-injection compared to the [^64^Cu]Cu-NGR monomer (3.33 ± 0.10%, 3.09 ± 0.20%, 2.32 ± 0.17%, and 1.79 ± 0.27% ID/g at 1-, 2-, 4-, and 24-h after the injection; respectively) [55]. The difference between the two probes may be attributed to the bivalency effect and the enlargement of the dimeric molecule. Although both studies published elevated in vivo T/M ratios for the radiolabelled dimeric agent, in the experiment of Israel et al., this was statistically not significant. Chen et al. registered a similar distribution pattern of [^64^Cu]Cu-NGR_1_ and [^64^Cu]Cu-NGR_2_ in the major organs with high hepatic and renal tracer uptake 1-, 2-, and 4-h post-injection and—*apart from the intestines*—slight tracer accumulation in the rest of the organs at all time points. Contrarily, in the study of Israel et al., the investigated tissues and organs exhibited higher uptake of the [^68^Ga]Ga-NGR monomer than the [^68^Ga]Ga-NGR dimer [47]. Although the exact reason why there was a difference regarding the uptake tendency of the monomer and dimer compounds between the two studies is not yet fully uncovered, we presuppose that the variabilities between the applied tumour types and the radiometals could be suggested as possible explanations. Furthermore, since in the experiment of Israel et al., the PET scans were acquired 50 min post-injection (10 min long), while Chen and coworkers performed in vivo imaging 1, 2, 4, (5 min long), and 24 h (10 min long) after tracer application, methodological differences may also partly explain the reason behind.

Comparing the ex vivo results recorded by Chen et al. and Israel et al.,—*regardless of the radiometal*—high hepatic tracer accretion was found for the dimeric compounds, which could be attributed to the elimination kinetics as well as the properties of the radiometal-chelator system and the NGR peptides. A brief overview of the physical properties of ^68^Ga and ^64^Cu is given in Table 3. In addition, compared to other ^68^Ga-labelled NGR probes ([^68^Ga]Ga-DOTA-NGR; [41]), both the ^64^Cu-labelled monomer and dimer exhibited enhanced liver uptake. While in the case of the A549 tumour-bearing mice, the hepatic [^68^Ga]Ga-DOTA-NGR concentration was only 0.2% ID/g at the early time points, the respective liver uptake values for [^64^Cu]Cu-DOTA-NGR_1_ and [^64^Cu]Cu-DOTA-NGR_2_ in receptor positive HT1080 tumour xenografts were 6.39 ± 0.41% ID/g and 6.49 ± 0.39% ID/g [41,55].

A year later, Li et al. published how they managed to synthesize [^64^Cu]Cu-labelled dimeric NGR peptide using sarcophagine cage (Sar) as a macrocyclic chelating agent [31]. Table 4 provides an overview of their study. Sarcophagine cage-based chelators complexed with ^64^Cu are stable enough to avoid the dissociation of the radiometal [31]. In comparison with other chelators, these types of bifunctional chelators ensure more rapid and selective conjugation with ^64^Cu^2+^ within an extensive range of pH values [81]. The performance of [^64^Cu]Cu-Sar-NGR_2_ was tested both in vitro and in vivo. The comparable receptor binding affinity of [^64^Cu]Cu-Sar-NGR_2_ (1.04 ± 0.26 nM) and the earlier synthesized dimeric NGR peptide indicates the augmentation of APN/CD13 receptor adhering capability induced by peptide multimerization [55].

As expected, significant tracer accretion was observed in HT1080 human fibrosarcoma cells with APN/CD13 overexpression (1.72 ± 0.24% 2 h post-incubation), while CD13-negative MCF-7 human breast adenocarcinoma cells did not exhibit meaningful [^64^Cu]Cu-Sar-NGR_2_ uptake (0.4 ± 0.03% 2 h after incubation). These in vitro findings further strengthen the specificity of the NGR homing sequence. Similarly to the in vivo results of Chen et al., the receptor positive HT1080 tumours were visualised with high contrast in comparison with the background; however, the APN/CD13 negative MCF-7 tumours showed only slight tracer accumulation in the bilateral tumour-bearing female athymic nude mice (HT1080 and MCF-7) [31,55]. The outstanding tumour retention even at 24-h post-tracer administration could be due to the prominent binding ability and the immense size of the probe. In agreement with the findings on [^64^Cu]Cu-DOTA-NGR_1_ and [^64^Cu]Cu-DOTA-NGR_2_ [55], elevated [^64^Cu]Cu-Sar-NGR_2_ uptake was registered at the early time points in the liver and in the kidneys, and—*except for the intestines*—minimal radioactivity was experienced in the other evaluated organs. High hepatic radioactivity may be underpinned by the expression of the APN/CD13 biomarker in the mouse liver as well as the prolonged hepatic retention of the radioisotope.

Markedly decreased tumour tracer concentration found upon the administration of the non-labelled blocking NGR peptide further validated the selectivity of this probe for the non-invasive target-specific assessment of APN/CD13 expression (in vivo: blocked vs. unblocked 1.05 ± 0.22%ID/g and 5.42 ± 0.27%ID/g; respectively; and ex vivo: blocked vs. unblocked 0.71 ± 0.13%ID/g and 2.98 ± 0.60%ID/g; respectively). Corresponding to the in vivo data, predominant [^64^Cu]Cu-Sar-NGR_2_ activity was noted ex vivo in the liver and in the kidneys; however, in the rest of the organs, the accumulation remained at a negligible level, and this is consistent with the organ distribution analysis of Chen et al. [31,55]. The detailed study of Li et al.—in correlation with that of Chen et al.—confirmed that ^64^Cu is suited for the accomplishment of the radiolabelling of NGR peptides and could be a valuable substitute for ^68^Ga. Furthermore, it draws attention to the applicability of chelators other than the conventionally applied ones.

Although ^64^Cu could be a viable alternative to ^68^Ga in the establishment of NGR motif containing PET probes, prolonged hepatic retention must be addressed. The slow dissociation of the radiometal from the chelating agent may lead to prolonged liver accumulation and subsequent difficulties regarding the precise characterisation of hepatic alterations [82]. Alternative chelators, including TETA (1,4,8,11-tetraazacyclotetradecane-N,N′,N″,N‴-tetraacetic acid), cross-bridged cyclam ligands, or sarcophagine (3,6,10,13,16,19-hexaazabicyclo [6.6.6]icosane), and the integration of linkers of adequate length, charge, hydrophilicity, and flexibility, may aid in overcoming the instability of the radiometal-chelate system, leading to the establishment of more appropriate pharmacokinetics [83,84,85,86,87,88]. Other differences between the imaging characteristics of [^68^Ga]Ga and [^64^Cu]Cu-labelled NGR probes may stem from the disparities between the two radiometals and the biochemical structure of the assessed NGR sequences [41]. Table 3 presents the comparison of the physical properties of ^68^Ga and ^64^Cu. In addition, taking the improved imaging behaviour of [^64^Cu]Cu-DOTA-NGR_2_ over [^64^Cu]Cu-DOTA-NGR_1_ into account, the construction of dimer peptide-based radiopharmaceuticals other than NGR is warranted.

### 2.3. Labelling with ^99m^Tc-Pertecnetate (^99m^Tc-Pertechnetate)

Besides PET radioisotopes, gamma emitting ^99m^Tc-pertecnetate (^99m^Tc) is also widely applicable for the radiolabelling of NGR peptides [6]. In the former study of Ma et al., the APN/CD13 binding specificity of [^99m^Tc]Tc-labelled monomeric and dimeric NGR derivatives was investigated in vivo in receptor positive HepG2 hepatoma-bearing female BALB/c nude mice using microSPECT [6]. Table 4 displays the most important characteristics of their study.

**Table 4 ijms-24-12675-t004:** Application of ^177^Lu, ^188^Re, ^213^Bi, ^64^Cu, and ^99m^Tc-labelled NGR-based peptide derivatives in cancer diagnostics.

Investigated Object	Investigated Phenomenon/Initiatives	Radiopharmaceutical	In Vitro and In Vivo Methods	Highlights	Reference
in vitro studies: HT1080 and HT29 cell lines in vivo studies: HT1080 and HT29 tumour-bearing athymic nude mice	in vitro properties, in vivo imaging capability, and APN/CD13 targeting ability	monomeric [^64^Cu]Cu-DOTA-NGR_1_ dimeric [^64^Cu]Cu-DOTA-NGR_2_	in vitro studies: in vitro stability, cell uptake and efflux studies, cell binding assay in vivo studies: in vivo static microPET imaging, ex vivo gamma counter measurements for radioactivity calculation, blocking experiments	both the dimer and the monomer specifically target APN/CD13 overexpressing HT1080 tumour xenografts, improved tumour accumulation and retention were found for the dimeric compound compared to the monomer, and dimeric [^64^Cu]Cu-DOTA-NGR_2_ is a better PET imaging probe for the in vivo visualisation of the APN/CD13 pattern	[55]
HT1080 and MCF-7 cells, bilater tumour-bearing female athymic nude mice (HT1080 and MCF-7)	synthesis process, in vitro, and in vivo performance evaluation	[^64^Cu]Cu-Sar-NGR_2_	in vitro cell binding assay, cell uptake, and efflux studies in vivo static microPET imaging, ex vivo uptake studies, and in vivo blocking examinations	Sarcophagine cage could be an alternative chelator for radiopharmaceutical development [^64^Cu]Cu-Sar-NGR_2_ is a useful diagnostic agent for the in vivo assessment of APN/CD13 expression	[31]
HepG2 cells, HepG2 hepatoma-bearing female BALB/c nude mice	APN/CD13 receptor binding specificity and comparison of monomer and dimer NGR derivatives	[^99m^Tc]Tc-NGR_1_ monomer [^99m^Tc]Tc-NGR_2_ dimer	in vitro cell binding assay and cell uptake studies, in vivo microSPECT imaging and blocking experiments, and ex vivo gamma counter measurements for organ distribution assessment	[^99m^Tc]Tc-NGR_2_ dimer presented more improved in vitro and in vivo properties than the monomer (binding potential, cell uptake, tumour accumulation and retention, pharmacokinetics), [^99m^Tc]Tc-NGR_2_ dimer is a valuable tracer for SPECT-based angiogenesis imaging	[6]
A549, PC-3, and OVCAR-3 cell lines and A549, PC-3, and OVCAR-3 tumour carrying athymic male nude mice	in vitro and in vivo angiogenesis imaging characteristic, organ distribution profile	[^99m^Tc]Tc-MAG3-PEG8-c(NGRyk)	in vitro cell binding assay, in vivo planar gamma camera imaging, ex vivo gamma counting, in vitro and in vivo receptor blocking studies	[^99m^Tc]Tc-MAG3-PEG8-c(NGRyk) appears to be an effective imaging probe, especially in cases of pulmonary malignancies	[49]
B16F10 and SKMEL28 cell lines and corresponding B16F10 and SKMEL28 xenografted nude mice	angiogenesis detection	[^99m^Tc]Tc-MAG3-PEG8-c(NGRyk)	in vitro internalization studies, ex vivo organ distribution, and blocking experiments	The human melanoma cells (SKMEL28) exhibited higher [^99m^Tc]Tc-MAG3-PEG8-c(NGRyk) uptake than the murine ones (B16F10)	[50]
in vitro studies: LLC and Colo 205 cells	in vitro studies: anti-angiogenic biological activity	in vitro studies: NGR-hPK5, wild-type hPK5	in vitro studies: cell proliferation assay (MTT assay), cell migration assay (transwell assay, and the wound healing assay) cord morphogenesis assay, CAM assay	NGR-hPK5 exerted a more prominent antiangiogenic effect and response than wild-type hPK5 (inhibition of proliferation, migration, and cord formation of vascular endothelial cells), and the addition of the NGR sequence to the antiangiogenic molecules could enhance their therapeutic potential	[89]
LLC carrying female C57BL/6J mice, athymic nude mice bearing Colo 205 tumours	diagnostic feasibility: tumour-targeting ability, in vivo and ex vivo organ distribution therapeutic effect: effect of NGR on the antitumour activity of hPK5, tumour growth inhibition	imaging: [^99m^Tc]Tc-hPK5, [^99 m^Tc]Tc-NGR-hPK5 therapy: hPK5, NGR-hPK5, hPK5/NGR-hPK5 in combination with cisplatin	imaging: in vivo planar imaging with a gamma camera; ex vivo radioactivity determination by gamma counting therapy: calliper measurements for tumour volume determination; CI index calculation	[^99 m^Tc]Tc-NGR-hPK5 showed greater tumour-homing ability than wild-type hPK5, NGR-hPK5 presented favourable anti-tumour effect, the NGR motif improved the anti-tumour potential of hPK5, and the addition of the NGR sequence to antiangiogenic molecules could enhance their therapeutic potential	[89]
cytotoxicity studies: B16F10 cell imaging: B16F10 tumour bearing C57BL6 mice	cytotoxicity studies: growth inhibition imaging: organ distribution pattern	cytotoxicity studies: cNGR-CLB complex, cNGR, CLB imaging: [^99m^Tc]Tc-HYNIC-CLB-c(NGR)	in vitro cytotoxicity studies: colorimetric MTT assay imaging: ex vivo biodistribution and blocking studies	the peptide-drug conjugate (cNGR-CLB) exhibited higher cytotoxicity than the peptide (cNGR) or the drug (CLB), so the conjugation of target-specific peptides with CLB or other chemotherapeutic agents could be used for the achievement of increased therapeutic efficacy	[34]
nude mice bilaterally bearing HT1080 tumours	radiochemical synthesis process, in vivo detection of APN/CD13 expression, and organ distribution pattern	^99m^TcO-N_3_S-PEG_2_-probestin	in vivo whole-body static planar imaging, microSPECT imaging, blocking experiments, and ex vivo radioactivity determination	^99m^TcO-N_3_S-PEG_2_-probestin shows APN selective uptake and high affinity APN inhibitor conjugates could be effectively used for the targeting of APN	[90]
HT1080 tumour-bearing CB17 SCID mice	^213^Bi: APN/CD13 positive tumour-homing ability, anti-tumour potential ^68^Ga: in vivo organ distribution, tumour targeting competence	[^213^Bi]Bi-DOTAGA-cKNGRE [^68^Ga]Ga-DOTAGA-cKNGRE	cancer treatment studies, ex vivo gamma counting with ^213^Bi, calliper measurements for tumour growth determination, body weight and tumour volume measurements, and in vivo MiniPET imaging with ^68^Ga	[^213^Bi]Bi-DOTAGA-cKNGRE is a potential weapon in the targeted treatment of APN/CD13 expressing fibrosarcoma [^68^Ga]Ga-DOTAGA-cKNGRE is a potent PET diagnostic probe for the detection of APN/CD13 overexpressing primary tumours and metastases	[30]
HT1080 cells, HT1080 xenografted BALB/c nude mice	in vivo diagnostic and therapeutic feasibility, in vitro inhibitory effect	[^188^Re]Re-NGR-VEGI	in vivo SPECT imaging and radiotherapy studies; ex vivo radioactivity determination by gamma counter; in vitro apoptosis assay by flow cytometry	[^188^Re]Re-NGR-VEGI can be efficiently used in theranostic settings for both the diagnostics and the targeted therapy of APN/CD13 tumours	[32]
B16F10 tumour-bearing C57BL6 mice	drug delivery, in vivo imaging properties, ex vivo organ biodistribution, and in vivo anti-tumour effect	([^177^Lu]Lu-DOTA-CNS-cNGR, ([^177^Lu]Lu-DOTA-cNGR, ([^177^Lu]Lu-DOTA-CNS)	in vivo uptake studies, blocking studies	[^177^Lu]Lu-DOTA-CNS-cNGR presented a higher tumour-to-background ratio than [^177^Lu]Lu-DOTA-CNS, CNS attached to target-specific molecules, can be used to design nanoprobes for targeted drug delivery	[35]

A549: human lung small cell carcinoma cell; APN/CD13: Aminopeptidase N; ^213^Bi: Bismuth-213; B16F10: murine melanoma cell line; CAM assay: Chick Embryo Chorioallantoic Membrane Assay; CLB: chlorambucil; CI index: combination index; Colo 205: human colon cancer cell line; CNS: carbon nanosphere; cNGR: cyclic NGR; cNKGRE: cyclic Lysine-Asparagine-Glycine-Arginine-Glutamic acid; ^64^Cu: Copper-64; DOTA: 1,4,7,10-teraazacyclododecane-N,N′,N″,N‴-teraacetic acid; DOTAGA: 1,4,7,10-tetrakis(carboxymethyl)-1,4,7,10-tetraazacyclododecane glutaric acid; ^68^Ga: Gallium-68; HepG2: human hepatocellular carcinoma cell line; HT1080: human fibrosarcoma cell line; HT29: human colon adenocarcinoma cell line; hPK5: human plasminogen kringle 5; HYNIC: hydrazinonicotinamide; LLC: Mouse Lewis lung carcinoma; ^177^Lu: Lutetium-177; MCF-7: human breast adenocarcinoma; MTT assay: cell proliferation assay; NGR: Asparagine-Glycine-Arginine; N_3_S: N,N-dimethylglycyl-l-lysinyl-l-cysteinylamide; PC-3: human prostate carcinoma cell line; PEG8: polyethylene glycol; ^188^Re: Rhenium-188; Sar: Sarcophagine cage; SCID: severe combined immunodeficient; SKMEL28: human melanoma cell line; SPECT: single photon emission computed tomography; ^99m^Tc: Technetium-99m; OVCAR-3: human ovarian adenocarcinoma cell line; VEGI: vascular endothelial growth inhibitor.

Due to the multimerization of the NGR motif, enhanced binding affinity and in vitro cellular uptake were reported for the NGR-containing dimer ([^99m^Tc]Tc-NGR_2_) compared to the monomer ([^99m^Tc]Tc-NGR_1_), which was similar to the previous results of Chen et al. and Israel et al. [47,55]. Earlier literature data also proved that the application of more than one targeting RGD unit notably increased the binding ability of RGD to the integrin α_v_β_3_ receptor via the polyvalency effect [70,82,91,92].

[^99m^Tc]Tc-NGR_2_ exhibited more enhanced HepG2 tumour uptake than the monomer, and this was consistent with the previous comparison analysis of [^64^Cu]Cu-labelled monomeric and dimeric NGR peptides [55]. The more favourable pharmacokinetics of [^99m^Tc]Tc-NGR_2_, such as rapid tumour uptake, prolonged washout from the tumour tissue, and lengthened circulation time, emphasises the advantages of peptide dimerization in the optimisation of the pharmacokinetic profile of the molecules. Overall, given the superiority of the dimer with regard to tumour uptake and tumour-to-non-target tissue contrast, better image quality could be obtained applying [^99m^Tc]Tc-NGR_2_ than [^99m^Tc]Tc-NGR_1_. Intriguingly, higher liver accumulation was determined for [^99m^Tc]Tc-NGR_2_ (15.87 ± 1.76%ID/g) than for the monomer (10.30 ± 1.32%ID/g), which was in accordance with the uptake values of [^68^Ga]Ga-NOTA-(NGR)_2_, [^64^Cu]Cu-DOTA-NGR_1_, [^64^Cu]Cu-DOTA-NGR_2_, and [^64^Cu]Cu-Sar-NGR_2_ [31,47,55]. Among others, this could presumably be due to the larger molecular size of the dimer or—as previously mentioned—the natural presence of APN/CD13 in the mouse liver. While the formerly detailed NGR radiotracers were eliminated mainly via the urinary system, in the case of these [^99m^Tc]Tc-labelled probes, both renal and hepatic excretion were present.

Another study with [^99m^Tc]Tc-labelled NGR peptides was conducted by Faintuch et al., who explored the in vitro and in vivo behaviour of pegylated cyclic NGRyk compounds appended with ^99m^Tc ([^99m^Tc]Tc-MAG3-PEG8-c(NGRyk)) in human lung carcinoma (A549), androgen-independent human prostate carcinoma (PC-3), and ovarian carcinoma (OVCAR-3) cell lines and corresponding mouse xenograft models (as presented in Table 4) [49]. To exploit the advantages of pegylation, such as protection against enzymatic degradation with related longer half-life and stability, the augmentation of hydrophilicity, and the reduction of immunogenicity, Faintuch et al. used polyethylene glycol (PEG8) as a spacer [93,94]. The highest uptake of the tracer was depicted in the kidneys both in the healthy Swiss (2.61 ± 1.02%ID/g 1 h post-injection) and the tumourous mice (2.36 ± 0.45, 3.40 ± 0.84, and 4.85 ± 2.15%ID/g for the PC-3, A549, and OVCAR-3 tumours, respectively, 1 h after injection). Identically, in a previous study of the same group dealing with [^99m^Tc]Tc-labelled cyclic pegylated pentapeptide NGRyk ([^99m^Tc]Tc-MAG3-PEG8-c(NGRyk)), the most prominent accumulation was also found in the kidneys, which was the major route of elimination (Table 4) [50]. The intestines, the liver, and the lungs exhibited relatively higher [^99m^Tc]Tc-MAG3-PEG8-c(NGRyk) accumulation in both studies [49,50]. The increased intestinal and hepatic activity indicates that—similarly to [^99m^Tc]Tc-NGR_1_ and [^99m^Tc]Tc-NGR_2—_the hepatobiliary system also plays a role in the elimination of [^99m^Tc]Tc-MAG3-PEG8-c(NGRyk) [6,49,50]. Although the in vitro cell binding assay indicated largely comparable binding ability for the pulmonary and ovarian cancer cells one hour post-incubation, the in vivo tumour accumulation was more prominent in OVCAR-3 tumour-bearing mice. We assume that this could be due to the difference between the spatial expression of the APN/CD13 receptors of the individual tumour cells and the entire tumour mass. Continuous changes in the receptor density of the tumours during tumour growth—induced by necrosis, chronic inflammation, or hypoxia—may also alter their radiopharmaceutical uptake. However, given that almost all non-target organs and tissues presented lower [^99m^Tc]Tc-MAG3-PEG8-c(NGRyk) activity in the A549 tumours than in the OVCAR-3 tumours, images with higher contrast and better T/M ratios were obtained in the case of the lung carcinoma bearing mice (T/M: 3.25 and 2.77 for the A549 and the OVCAR-3 tumours, respectively). Likewise, high T/M ratios were registered for [^99m^Tc]Tc-MAG3-PEG8-c(NGRyk) in the B16F10 (5.0) and SKMEL28 (approximately 3) melanoma tumourous mice [50]. Comparing the uptakes of the three assessed tumours, the lowest values were registered in the case of the PC-3 tumour carrying mice (0.36 ± 0.11, 0.65 ± 0.14 and 0.97 ± 0.38%ID/g for the PC-3, A549, and OVCAR-3 tumours; respectively). As reported by *Oliveira and co-workers*, the radioactivities of SKMEL28 human melanoma (1.07 ± 0.23%ID/g) and B16F10 mouse melanoma (0.85 ± 0.34%ID/g) tumour xenografts were almost identical to those of the OVCAR-3 tumours (0.97 ± 0.38%ID/g) (demonstrated in Table 4) [50].

Since the cyclic CNGRC motif may have the potential to enhance tumour homing, Jiang et al. performed in vivo planar imaging as well as organ distribution studies using the [^99m^Tc]Tc-labelled derivatives of a wild-type endogenous angiogenic inhibitor—*named human plasminogen kringle 5 (hPK5)*—and its CNGRC-attached match (NGR-hPK5) in Mouse Lewis lung carcinoma (LLC) carrying female C57BL/6J mice (Table 4) [89,95,96]. To test the effect of cyclic CNGRC on the anti-tumour activity of hPK5, mouse models of LLC and human colorectal adenocarcinoma (Colo 205) were treated with hPK5/NGR-hPK5 and hPK5/NGR-hPK5 with cisplatin.

The 3-times higher tumour accumulation of [^99m^Tc]Tc-labelled NGR-hPK5 compared to the wild-type molecule reflects the enhancement of tumour targeting capability induced by the incorporation of the NGR sequence. Due to its selective LLC tumour accumulation without nonspecific uptake in the normal tissues and higher tumour-to-non-target ratios, it can be concluded that the performance of the NGR-linked derivative exceeds that of the wild-type one in terms of tumour visualisation.

Based on the fact that more significant tumour growth inhibition was achieved with the NGR-modified compound both in the LLC and Colo 205 tumour-bearing mice, NGR peptides could be pioneering in the targeted delivery of anti-cancer drugs. Moreover, cisplatin administration in combination with NGR-attached hPK5 led to increased suppression of tumour neovascularization compared to the application of hPK5 alone plus cisplatin. This strengthened the role of the NGR motif in increasing the anti-tumour effect of the peptide. Accordingly, immunofluorescence staining using CD31 antibody also proved that the microvessel density of the LLC tumours was reduced to a greater extent upon NGR-hPK5 administration in comparison with the wild type hPK5. Examining the cytotoxicity of the NGR peptide connected to DNA alkylating nitrogen mustard chlorambucil (CLB), Vats et al. also stated that the attachment of target-specific peptides to chemotherapeutic agents could enhance the anti-tumour effect [34]. They registered enhanced cytotoxicity and tumour cell growth inhibition in B16F10 melanoma cells using the cNGR-CLB complex relative to the NGR peptide or the drug alone. In addition, the cNGR-CLB complex was furnished with ^99m^Tc ([^99m^Tc]Tc-HYNIC-CLB-c(NGR); HYNIC: hydrazinonicotinamide), and biodistribution studies were performed in B16F10 melanoma tumour-bearing C57BL6 mice. The rapid tumour uptake (maximum at 30 min post-injection), suitable T/M/tumour-to-blood ratios, and prompt clearance from the non-target normal tissues support the diagnostic value of [^99m^Tc]Tc-HYNIC-CLB-c(NGR) as well. Their study is detailed in Table 4.

In the study of Pathuri et al., the APN/CD13 inhibitor probestin radiolabelled with ^99m^Tc was used for the in vivo detection of APN/CD13 expression in nude mice transplanted with HT1080 human fibrosarcoma tumour cells (as displayed in Table 4) [90,97]. The complexation of the new N(3)S-chelated probestin conjugate with ^99m^Tc was achieved by transmetallation from ^99m^Tc(V)-gluconate [98]. To track the pharmacokinetics and the biodistribution of ^99m^TcO-N_3_S-PEG_2_-probestin, nude mice bilaterally bearing HT1080 tumours in their suprascapular region were generated for microSPECT imaging and biodistribution studies. Based on the definitive tumour visualisation ensured by high tumour uptake values (2.88 ± 0.64 %ID/g 1 h post-injection), appropriate tumour-to-blood/muscle ratios, and the rapid clearance of ^99m^TcO-N_3_S-PEG_2_-probestin from the circulation (0.60 ± 0.33 %ID/g 1 h post-injection), radiolabelled APN/CD13 specific inhibitors also seem to be precious candidates for the non-invasive identification of receptor expression. The reversible receptor binding of the vector favours early phase imaging and the application of isotopes with short longevity, which is another clear advantage of ^99m^TcO-N_3_S-PEG_2_-probestin.

Given the target specificity of the examined [^99m^Tc]Tc-labelled APN/CD13 targeting probes as well as the easy accessibility and applicability of the radiometal, ^99m^Tc serves as a valuable alternative for the radiolabelling of NGR ligands. Further, these findings collectively demonstrate that the therapeutic efficacy of anti-cancer drugs could be augmented by attaching them to angiogenesis-specific peptide ligands; moreover, this could be a novel pharmacological approach to battling cancer. Table 4 summarises the preclinical studies with [^99m^Tc]Tc-labelled radiotracers with selective APN/CD13 homing capability.

### 2.4. Labelling with Bismuth-213 (^213^Bi)

Given that the high linear energy transfer and short tissue penetration range of the alpha emitter Bismuth-213 (^213^Bi) ensure selective tumour toxicity while sparing the healthy tissues nearby, it is embraced as a promising isotope in targeted radionuclide therapy (TRNT) [99,100]. Hence, the evaluation of [^213^Bi]Bi-labelled radiotracers in therapeutic settings is the central focus of current oncological research [101]. Evaluating the tumour homing ability and the anti-tumour potential of [^213^Bi]Bi-DOTAGA-cKNGRE, Képes et al. assumed for the first time the usefulness of [^213^Bi]Bi-labelled molecular probes in angiogenesis-targeted cancer treatment (DOTAGA: 1,4,7,10-tetrakis(carboxymethyl)-1,4,7,10-tetraazacyclododecane glutaric acid; demonstrated in Table 4) [30]. CB17 severe combined immunodeficient (SCID) mice bearing HT1080 tumours were intraperitoneally injected with 4.68 ± 0.10 MBq of [^213^Bi]Bi-DOTAGA-cKNGRE to test the effect of the agent on the body weight and the tumour volume of the animals. The tumour volumes of the treatment-naïve animals (63.00 ± 11.53 mm^3^, 155.43 ± 20.47 mm^3^, and 204.43 ± 28.36 mm^3^ on days 9, 10, and 12; respectively) and the treated tumourous animals (35.36 ± 7.57 mm^3^, 58.33 ± 14.36 mm^3^, and 112.50 ± 18.42 mm^3^ on days 9, 10, and 12, respectively) showed significant differences, which led to the conclusion that [^213^Bi]Bi-DOTAGA-cKNGRE exerted a meaningful anti-tumour effect.

Similarly to the [^99m^Tc]Tc-labelled NGR derivatives [6,49,50], the role of the hepatobiliary system could be assumed in the overall elimination of [^213^Bi]Bi-DOTAGA-cKNGRE. Suitable image contrast/quality obtained at 90 min post-tracer administration suggest that the diagnostic value of the gamma rays of the radiometal could also be exploited.

[^68^Ga]Ga-DOTAGA-cKNGRE—used for the identification of the HT1080 tumours—showed significant tumour uptake coupled with faint background activity (tumour SUV_mean_ and SUV_max_: 0.37 ± 0.09 and 0.86 ± 0.14 90 min post-injection, respectively). This was similar to the results of previous studies with different c(NGR) derivatives, such as [^68^Ga]Ga-DOTA-c(NGR)_2_ [44] and [^68^Ga]Ga-NOTA-c(NGR) [33,42]. 

These initial results with ^213^Bi highlight the importance of the proposal of radiolabelled, target-specific therapeutic probes that could circumvent the adverse effects derived from currently applied oncological treatments. Furthermore, the above mentioned findings create the opportunity for the seamless employment of the [^68^Ga]Ga-DOTAGA-cKNGRE/[^213^Bi]Bi-DOTAGA-cKNGRE radiotracer pair as a theranostic agent in APN/CD13 positive tumour diagnostics, therapy, and therapeutic response assessment.

### 2.5. Labelling with Rhenium-188 (^188^Re)

In a bid to battle against the adverse health ramifications of traditional radiotherapy, intensive focus needs to be placed on the introduction of targeted therapeutic approaches into human patient care. Due to the tumour selectivity of NGR tripeptides, they seem to be useful carriers of chemotherapeutic drugs, proapoptotic molecules, or tumour necrosis factor into the tumour tissue or in close proximity [102,103,104]. Owing to the therapeutic β^-^ particles (2.12 MeV) of Rhenium-188 (^188^Re), along with its gamma co-emission (155 keV), it is applicable for the labelling procedure of NGR molecules not only in imaging but also in radiotherapeutic settings [105,106].

Based on literature data, radiolabelled NGR-drug conjugates in complexation with anti-angiogenic vascular endothelial growth inhibitor (VEGI) could be effectively used in theranostic applications [32]. Ma et al. developed a [^188^Re]Re-labelled NGR-VEGI fusion protein as a possible theranostic agent for the SPECT diagnostics and the targeted therapy of HT1080 xenografted female BALB/c nude mice. Table 4 provides an overview of the major characteristics of the in vivo study of Ma et al. The imaging capability of [^188^Re]Re-NGR-VEGI was confirmed by the satisfactory tumour-to-background tissue ratios experienced upon SPECT image interpretation and biodistribution data analysis.

Taking the considerable in vivo tumour growth inhibition—induced by the administration of 18.5 MBq of [^188^Re]Re-NGR-VEGI into account—it may emerge as a potentially new approach to cancer therapy. The anti-tumour activity of [^188^Re]Re-NGR-VEGI was more prominent than that of the NGR peptide, or NGR-VEGI. The successful cytotoxic effect of the probe was further confirmed by in vitro therapeutic studies presenting notably higher tumour cell necrosis and apoptosis in the case of [^188^Re]Re-NGR-VEGI than regarding the vector control, NGR peptide, VEGI protein, and the unlabelled NGR-VEGI protein. The lack of therapy associated systemic toxicity is another advantage of this agent.

Consequently, ^188^Re could not only be used for the radiolabelling of NGR derivatives but also for the construction of NGR ligand containing molecular probes for the simultaneous imaging and targeted therapy of APN/CD13 overexpressing neoplasms.

### 2.6. Labelling with Lutetium-177 (^177^Lu)

To explore the potential of nanoparticles in simultaneous drug delivery, in vivo imaging, and anti-tumour therapy, Vats et al. produced a tumour vasculature targeting radionanosphere system [35]. Due to its therapeutic β emission and imageable gamma photons, Lutetium-177 (^177^Lu) was used for the radiolabelling of the novel nanoplatform made up of carbon nanospheres (CNS) attached to the G_3_-cNGR peptide (seen in Table 4) [35,107].

To assess the imaging properties of this nanosphere-peptide conjugate ([^177^Lu]Lu-DOTA-CNS-cNGR), its organ distribution pattern and tumour affinity were compared with those of the radiolabelled NGR peptide without the nanosphere ([^177^Lu]Lu-DOTA-cNGR) as well as the peptide-free [^177^Lu]Lu-labelled CNS ([^177^Lu]Lu-DOTA-CNS) using B16F10 melanoma tumour-bearing C57BL6 mice. The more enhanced tumour uptake of [^177^Lu]Lu-DOTA-CNS-cNGR coupled with a more suitable tumour-to-background contrast compared with the other probes may allow us to draw the conclusion that the application of peptide-nanosphere complexes may herald a new era in target-specific nanomedicine.

Therefore, the use of radiolabelled nanospheres in conjunction with chemotherapeutic agents and target affine vectors grants the unique opportunity for the achievement of tumour-specific accumulation and therapeutic effect at the disease site without untoward non-target uptake and toxicity.

## 3. Radiolabelled NGR Peptides beyond Cancer Imaging

### 3.1. Radiolabelled NGR Peptides for Cardiac Imaging

Hendrikx et al. applied for the first time a cNGR homing sequence furnished with Indium-111 (^111^In) for the imaging of APN/CD13 in infarcted myocardial tissue [108]. Dual-isotope microSPECT using [^111^In]In-DOTA-cNGR combined with ^99m^Tc-sestamibi was performed in Swiss mice 7 days after myocardial ischaemia induction by the ligation of the left anterior descending coronary artery (LAD) (MI mice). MicroSPECT imaging verified that [^111^In]In-DOTA-cNGR is eliminated by the kidneys and is rapidly washed out of the intact myocardium. Markedly higher [^111^In]In-DOTA-cNGR accumulation was experienced in the underperfused infarcted myocardial regions or at the border zone with diminished ^99m^Tc-sestamibi uptake. Ex vivo gamma counting was carried out both in healthy control and MI mice for the qualitative measurement of myocardial [^111^In]In-DOTA-cNGR uptake and for the assessment of the normal distribution pattern. This study draws attention to the fact that radiolabelled NGR probes could be used not only for the imaging of cancer-related angiogenic processes but also for the imaging of APN/CD13 expression in perfusion-deficient myocardial regions. In addition, NGR-based molecules have also been extensively investigated in the detection of myocardial angiogenesis. APN/CD13 affine nanoparticles, including cNGR peptides targeting quantum dots and liposomes, were proposed for the imaging of highly angiogenic myocardial regions with APN/CD13 overexpression [109,110].

### 3.2. Radiolabelled NGR Peptides for Ischaemia-Reperfusion Imaging

Another notable application of NGR tripeptide motif containing radiopharmaceuticals is the detection of ischemia-reperfusion (IR) induced angiogenic alterations. In the study of Farkasinszky et al., the performance of [^68^Ga]Ga-labelled cNGR peptide was evaluated in the identification of the temporal changes of the APN/CD13 pattern in a rat model of surgically-induced diabetic retinopathy (DR) [111]. [^68^Ga]Ga-NOTA-cNGR was subjected to in vivo PET studies in male Fischer-344 rats with DR induced by the ligation of the left bulbus oculi (nervus opticus, arteria ophthalmica, and arteria ciliares) and in control animals as well. Considerably higher [^68^Ga]Ga-NOTA-cNGR uptake of the ligated left eye was measured at all time points (1, 3, 7, and 10 days after surgery) compared to the radioactivity of the eyes of the control group, which proves the applicability of the tracer in the detection of IR-mediated APN/CD13 expression. Farkasinszky et al. concluded that the role of cellular, vascular, and physiological autoregulation mechanisms, including perfusion changes in the optic neurons or smooth muscle depolarisation, as well as alterations in the permeability of the blood-retinal barrier induced by APN/CD13-related extracellular matrix degradation, could be responsible for the significant [^68^Ga]Ga-NOTA-cNGR accretion experienced in the right bulbs (non-ligated) of the surgical cohort 10 days after IR induction [112,113,114,115].

As these studies confirmed the diagnostic feasibility of NGR radiotracers in areas other than cancer-related angiogenesis, we expect to see more NGR-based radiopharmaceuticals being developed in the forseeable future in a wider range of non-cancerous applications.

### 3.3. Radiolabelled NGR Peptides for Therapeutic Applications

Furthermore, Pastorino et al. aimed to test the applicability of NGR peptides in therapeutic approaches [116]. To enhance the chemotherapeutic effect of doxorubicin (DXR), NGR peptide was attached to the surface of liposomal DXR (NGR peptide-targeted liposomal doxorubicin/NGR-SL[DXR]) and was applied to treat orthotopic neuroblastoma (NB) tumours generated in female SCID mice. NGR-coated liposomes were characterised by prolonged blood circulation time, which plays a crucial role in exerting their therapeutic effect. Biodistribution studies carried out 2,—12—and 24-h post-injection of [^3^H]H-CHE-labelled targeted NGR-SL[DXR] or non-targeted liposomal doxorubicin (SL[DXR]) indicated 10-fold higher tumour accumulation for NGR-SL[DXR] than for SL[DXR] 24 h after the injection. Based on this result, we may state that the addition of the NGR motif to SL[DXR] enhances its tumour-homing capability and therapeutic potential. Apart from the spleen, where significantly higher NGR-SL[DXR] uptake was found, the radioactivity of the two molecules did not differ in the investigated organs. We suppose that the increased lienal concentration of the targeted liposome could be caused by its relatively larger molecular size and related greater uptake by the reticuloendothelial cells. The following results of the in vivo therapeutic studies definitively strengthened the potential of NGR tripeptide in therapeutic settings: Rapid tumour volume reduction or even complete tumour remission, reduction of blood vessel density, and inhibition of metastatic spread were experienced in mice treated with NGR-SL[DXR] compared to the control cohorts receiving HEPES buffer (4-(2-hydroxyethyl)-1-piperazineethanesulfonic acid, untreated group), mismatched peptide ARA-targeted SL[DXR] (ARA-SL[DXR]) or NGR peptide-targeted liposome (NGR-SL) without DXR. In a similar manner, the histopathological findings—derived from the treated mice—such as disruption of tumour vasculature, suppression of blood vessel density, and apoptosis of the endothelial and tumour cells also support the feasibility of NGR in the development of targeted drug-delivery systems. Moreover, based on previous studies by Zhang et al. and Palmowski et al., ultrasmall superparamagnetic iron oxide USPIO nanoparticles and poly(butyl cyanoacrylate) PBCA-based microbubbles may emerge as tumour selective nanotheranostics that could be suitable for angiogenesis targeting, magnetic resonance (MR), ultrasound (US), and NIRF-based imaging, as well as for therapeutic purposes, due to their facile attachment to anti-cancer drugs and their favourable surface chemistry [117,118,119].

Together with the above remarked conclusions, NGR-based chemotherapeutic drugs targeting both the tumour vasculature and the tumour itself could open the way towards a novel form of anti-cancer treatment that may overcome the limitations of conventional oncotherapy. Although NGR-containing targeted therapeutics are still in their early stages of application, they potentially lay the groundwork for the establishment of a more specific way of treating cancer, eventually leading to the establishment of individualised anti-tumour therapy.

## 4. Future Perspectives, Closing Remarks

Radiolabelled NGR peptides are of particular value in the target-specific imaging of APN/CD13 positive tumour masses and related angiogenic processes. The target selectivity and in vivo imaging properties of NGR containing radiotracers capitalise on the relevance of tumour-homing NGR ligands for the design of APN/CD13 affine PET and SPECT imaging probes. This would not only broaden the horizons of imaging in the field of cancer medicine; however, it would also contribute to a deeper understanding of the mechanisms underlying disease pathogenesis.

Given the three times higher receptor targeting capability of NGR in comparison with RGD, angiogenesis imaging using NGR-based molecular vectors may lead to images with better quality [22]. This would ensure the seamless transition of NGR compounds into routine clinical usage, contributing to the wider spread of translational research. Furthermore, the co-development of APN/CD13 targeting diagnostic and therapeutic probes simultaneously allows for the identification of those patient populations who might benefit from anti-angiogenic targeted therapy as well as therapeutic response assessment.

Considering that tumour cells use various pathways for their survival, the establishment of multi-targeted diagnostic probes has attracted the interest of many investigators in oncological research fields. For example, in the study of Gai et al., a [^68^Ga]Ga-labelled heterodimeric tracer with NGR and RGD motifs ([^68^Ga]Ga-NGR-RGD) showed excellent in vivo imaging performance in mouse models of different breast cancers and pulmonary metastases [120]. This encouraging result provides the foundation for the development of NGR and RGD peptide-based dual-targeting molecular probes both for cancer-related angiogenesis imaging and targeted radiotherapy.

Although the investigated NGR probes display rapid tumour uptake together with slow washout, negligible off-target accumulation, and fast elimination from the blood, the fulfilment of chemical modifications to further optimize pharmacokinetics may be part of future work.

## Figures and Tables

**Figure 1 ijms-24-12675-f001:**
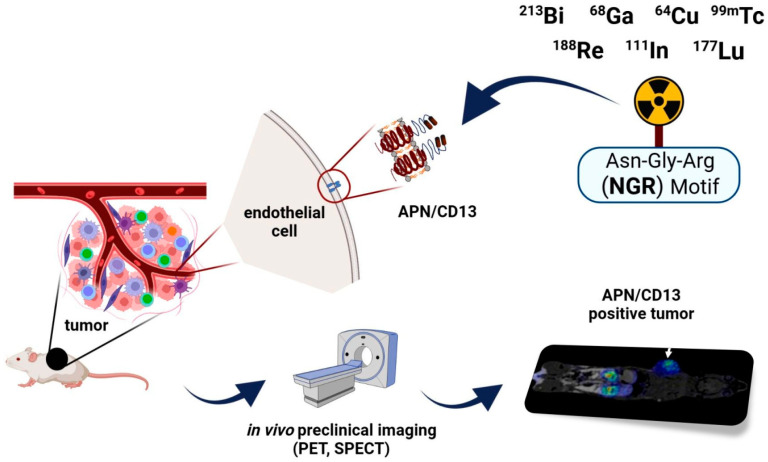
Overview of positron emission tomography (PET) and single-photon emission computed tomography (SPECT) radioisotopes applied for the labelling of APN/CD13 specific NGR motif including Gallium-68 (^68^Ga), Copper-64 (^64^Cu), Technetium-99m (^99m^Tc), Lutetium-177 (^177^Lu), Rhenium-188 (^188^Re), or Bismuth-213 (^213^Bi) in the imaging of cancer-associated neo-angiogenesis. APN/CD13: aminopeptidase N, NGR: asparagine-glycine-arginine. The figure was edited based on the free online version of biorender.com (accessed on 9 May 2023).

**Figure 2 ijms-24-12675-f002:**
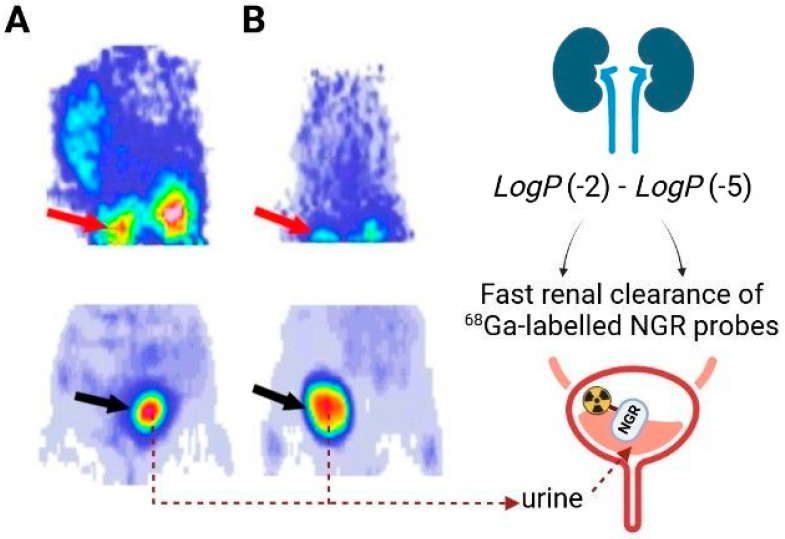
In vivo assessment of [^68^Ga]Ga-NOTA-c(NGR) (**A**) and [^68^Ga]Ga-NODAGA-YEVGHRC (**B**) accumulation in healthy control mice. Representative decay-corrected coronal PET images were obtained 90 min after the intravenous injection of the radiotracers. Red arrows: kidney; balck arrows: urinary bladder (urine). c(NGR): cyclic asparaginyl-glycyl-arginine; ^68^Ga: Gallium-68; NODAGA: 1,4,7-triazacyclononane,1-glutaric acid-4,7-acetic acid; NOTA: 1,4,7-triazacyclononane-triacetic acid; PET: positron emission tomography; YEVGHRC: tyrosyl-glutamyl-valyl-glycyl-histidyl-arginyl-cysteine [40] with permission and the figure was edited based on the free online version of biorender.com.

**Table 1 ijms-24-12675-t001:** Overview of preclinical studies with [^68^Ga]Ga-labelled APN/CD13 targeting NGR derivatives.

Investigated Object	Investigated Phenomenon/Initiatives	Radiopharmaceutical	In Vitro and In Vivo Methods	Highlights	Reference
in vitro: HT1080 and HT29 cells in vivo: female nude BALB/c mice bearing HT1080 or HT29 tumours	in vitro properties, in vivo diagnostic potential, and APN/CD13 positive tumour-homing capability	[^68^Ga]Ga-NOTA-G_3_-NGR	in vitro: stability, lipophilicity, cell-based competitive assay (binding affinity and specificity), tumour cell uptake and efflux in vivo: microPET/CT acquisition, blocking studies, and ex vivo gamma counter-based radioactivity determination	[^68^Ga]Ga-NOTA-G_3_-NGR is applicable in the diagnostics of CD13-targeted tumour angiogenesis, [^68^Ga]Ga-NOTA-G_3_-NGR is a valuable CD13-affine PET probe	[36]
in vitro: HT1080 and HT29 cells in vivo: female nude BALB/c mice xenografted with HT1080 and HT29	in vitro and in vivo comparative characterisation and comparison of angiogenesis imaging ability	[^68^Ga]Ga-NOTA-G_3_-NGR_2_	in vitro: stability, competitive cell binding assay (binding affinity and specificity), in vivo: microPET studies, ex vivo biodistribution	[^68^Ga]Ga-NOTA-G_3_-NGR_2_ is useful for angiogenesis imaging in preclinical HT1080 tumour models [^68^Ga]Ga-NOTA-G_3_-NGR_2_ seems to be valuable for the PET diagnostics of patients with fibrosarcoma [^68^Ga]Ga-NOTA-G_3_-NGR_2_ could be used to track CD13-targeted treatment response in a non-invasive manner	[37]
adult male Fischer-344 rats bearing orthotopic and heterotopic transplanted Ne/De tumours	radiochemical synthesis, APN/CD13 positive primer tumour targeting competence, metastasis identification, and comparative characterisation of the imaging properties of NGR and RGD peptides	[^68^Ga]Ga-NOTA-c(NGR)	in vitro: stability in vivo: whole-body miniPET examinations, ex vivo gamma counting, in vivo and ex vivo blocking studies	[^68^Ga]Ga-NOTA-c(NGR) selectively binds to APN/CD13 positive ortho- and heterotopic transplanted NeDe tumours [^68^Ga]Ga-NOTA-c(NGR) is a promising radionuclide for the in vivo imaging of APN/CD13 overexpressing primary tumours and related metastatic lesions	[33]
chemically induced He/De and Ne/De tumour-bearing Fischer-344 rats	evaluation and comparison of APN/CD13 selectivity	[^68^Ga]Ga-NOTA-c(NGR) [^68^Ga]Ga -NODAGA-c(NGR) [^68^Ga]Ga-NODAGA-c(NGR) (MG1) [^68^Ga]Ga -NODAGA-c(NGR) (MG2)	in vitro: stability in vivo: PET/MRI imaging, ex vivo uptake pattern, in vivo and ex vivo blocking studies	APN/CD13 is a valuable biomarker for PET diagnostic purposes, the careful choice of the ^68^Ga-NGR radiotracer is pivotal to correctly representing tumour-related angiogenesis and to following treatment response	[38]
chemically induced He/De tumour carrying male Fischer-344 rats	detection of the temporal changes of APN/CD13 expression, hypoxia imaging	[^68^Ga]Ga-NOTA-c(NGR)	in vivo: whole-body PET/MRI acquisition, blocking experiments	[^68^Ga]Ga-NOTA-c(NGR) has the potential to identify the temporal changes of hypoxic regions and the presence of APN/CD13 in hepatocellular carcinoma xenografts	[39]
in vitro: B16F10 cells in vivo: C57BL/6 mice bearing B16F10 tumours	imaging of neo-vascularisation and performance evaluation of angiogenesis-selective radiolabelled peptide porbes APN/CD13 selective LN peptide (YEVGHRC) VEGFR-1 selective (APRPG)	[^68^Ga]Ga-NODAGA-YEVGHRC [^68^Ga]Ga-NODAGA-APRPG-NH_2_ [^68^Ga]Ga-NODAGA-APRPG-COOH [^68^Ga]Ga-NOTA-cNGR (reference compound)	in vitro: stability determination in vivo: static microPET scans, ex vivo radioactivity determination	[^68^Ga]Ga-NODAGA-YEVGHRC, [^68^Ga]Ga-NODAGA-APRPG-NH_2_, and [^68^Ga]Ga-NODAGA-APRPG-COOH exhibited specific uptake in B16F10 tumours, [^68^Ga]Ga-NODAGA-APRPG-COOH displayed similar characteristics to those of [^68^Ga]Ga-NOTA-cNGR	[40]
in vitro: A549 and MDA-MB231 cells in vivo: A549 and MDA-MB231 carcinoma bearing female Balb/c nude mice	synthesis and evaluation of the efficacy of the detection of APN/CD13 and APN/CD13 expressing tumours	[^68^Ga]Ga-DOTA-NGR	in vitro: stability, cell uptake and binding studies in vivo: in vivo static and dynamic microPET acquisition, ex vivo organ distribution, in vivo blocking examinations	[^68^Ga]Ga-DOTA-NGR shows high affinity for APN/CD13 receptors both in vitro and in vivo*,* [^68^Ga]Ga-DOTA-NGR could be a promising diagnostic probe for the imaging of APN/CD13 expressing tumours and vasculature	[41]
in vivo: female Fischer-344 rats bearing primary and metastatic mesoblastic nephroma (Ne/De) tumours	detection of serially transplanted metastases and assessment of the change in APN/CD13 and α_v_β_3_ integrin receptor expression	[^68^Ga]Ga-NOTA-c(NGR)	in vivo: static microPET imaging	[^68^Ga]Ga-NOTA-c(NGR) is a useful tracer to detect primary Ne/De tumours and their thoracic parathymic lymph node metastases, the upregulation of the presence of APN/CD13 during serial metastasis transplantation projects increased malignancy	[42]
in vitro: HT1080 and HT29 cell lines in vivo: Female Balb/c nude mice with HT1080 and HT29 xenografts	imaging of APN/CD13 overexpressing tumours, comparative performance evaluation, and role of NRP-1	[^68^Ga]Ga-DOTA-iNGR with CendR (R/KXXR/K) penetrating motif, [^68^Ga]Ga-DOTA-NGR	in vitro: stability, cell binding affinity studies, cell uptake analysis, cell blocking studies with neutralising NRP-1 antibody in vivo: microPET imaging, ex vivo organ uptake studies, in vivo blocking studies with unlabelled cyclic NGR peptide or neutralising NRP-1 antibody	the use of the CendR motif led to higher tumour accumulation and increased tumour retention, indicating its capability to enhance the in vivo behaviour of NGR peptides	[43]
in vitro ES2 and SKOV3 cell lines in vivo: ES2 and SKOV3 tumourous female Balb/c nude mice	investigation of imaging efficacy	[^68^Ga]Ga-DOTA-c(NGR)_2_	in vitro: cell binding (affinity and specificity) assay; cell-based competitive cell binding assay; in vivo: static microPET examinations; ex vivo gamma counter-based measurements; in vivo receptor blocking studies	[^68^Ga]Ga-DOTA-c(NGR)_2_ demonstrated specific binding to APN/CD13 both in vitro and in vivo in ovarian tumour models, and [^68^Ga]Ga-DOTA-c(NGR)_2_ could be effectively applied in the PET diagnostics of APN/CD13 overexpressing tumours	[44]
in vitro: B16F10 and HT1080 cell lines in vivo: B16F10 tumour-bearing C57BL6 mice, athymic nude mice bearing HT-1080 tumours	in vivo imaging behaviour, tumour targeting ability	[^68^Ga]Ga-HBED-CC-c(NGR)	in vitro: cell uptake studies in vivo: biodistribution assessment, blocking experiments	[^68^Ga]Ga-HBED-CC-c(NGR) shows APN/CD13 target specificity and could be a promising diagnostic radiopharmaceutical for the imaging of receptor positive malignancies, chelator HBED-CC seems feasible for the complexation with imaging vectors other than NGR, and for the design of peptide-based radiotraces	[45]
in vitro: HT1080 tumour cells in vivo: nude mice xenografted with HT1080 tumours	exploring the influence of chelators on chemical properties, target selectivity, and distribution pattern	[^68^Ga]Ga-DOTAGA-c(NGR) [^68^Ga]Ga-NODAGA-c(NGR) [^68^Ga]Ga-HBED-CC-c(NGR)	in vitro: cellular uptake studies, uptake inhibition studies, efflux studies in vivo: ex vivo biodistribution studies, blocking studies	[^68^Ga]Ga-DOTAGA-c(NGR), [^68^Ga]Ga-NODAGA-c(NGR), and [^68^Ga]Ga-HBED-CC-c(NGR) are applicable molecular agents for the in vivo identification of tumours with APN/CD13 expression, [^68^Ga]Ga-NODAGA-c(NGR) exhibited more improved imaging performance compared to the other two probes	[46]
in vitro A549, SKHep-1, MDA-MB-231 cell lines in vivo: *CD1-Foxn1nu*-mice bearing A549 or SKHep-1 tumours	evaluation of the effect of NGR dimerisation and comparison of the target specificity of NGR monomer and dimer	[^68^Ga]Ga-NODAGA-NGR (NGR monomer) [^68^Ga]Ga-NOTA-(NGR)_2_ (NGR dimer)	in vitro: cellular uptake studies; in vivo: microPET imaging; in vivo uptake inhibition with APN/CD13 inhibitor bestatin; ex vivo accumulation profile	the monomer and the dimer tracers showed similar imaging behviour; both [^68^Ga]Ga-NODAGA-NGR and [^68^Ga]Ga-NOTA-(NGR)_2_ are favourable to detect APN/CD13 positive tumours without the superiority of the dimer to the monomer	[47]

A549: human lung small cell carcinoma cell; APN/CD13: Aminopeptidase N; B16F10: murine melanoma cell; CendR: C-end rule; c(NGR): cyclic NGR; ES2: human ovarian cancer cell; [^68^Ga]Ga: gallium-68; G_3_-NGR: (Gly_3_-CNGRC); DOTA: 1,4,7,10-teraazacyclododecane-N,N′,N″,N‴-teraacetic acid; DOTAGA: 1,4,7,10-tetrakis(carboxymethyl)-1,4,7,10-tetraazacyclododecane glutaric acid; [^18^F]F: Fluorine-18; HBED-CC: *N*,*N*′-bis-[2-hydroxy-5-(carboxyethyl)benzyl]ethylenediamine-*N*,*N*′-diacetic acid; He/De: hepatocellular carcinoma; HT1080: human fibrosarcoma cell line; HT29: human colon adenocarcinoma cell line; iNGR: internalising NGR; MDA-MB-231: human breast carcinoma cell line; Ne/De: mesoblastic nephroma tumours; NGR: Asparagine-Glycine-Arginine; NODAGA: 1,4,7-triazacyclononane,1-glutaric acid-4,7-acetic acid; NOTA: 1,4,7-triazacyclononane-triacetic acid; NRP-1: Neuropilin-1; PET/CT: positron emission tomography/computed tomography; PET/MRI: positron emission tomography/magnetic resonance imaging; RGD: arginine-glycine-aspartate; SKHep-1: human liver adenocarcinoma cell; SKOV3: human ovarian cancer cell line; YEVGHRC: tyrosyl-glutamyl-valyl-glycyl-histidyl-arginyl-cysteine.

**Table 2 ijms-24-12675-t002:** *LogP* values of the ^68^Ga-labelled tracers discussed in this review.

Radiopharmaceutical	*LogP*
[^68^Ga]Ga-NOTA-c(NGR)	−2.77 ± 0.12
[^68^Ga]Ga-NODAGA-c(NGR)	−4.07 ± 0.13
[^68^Ga]Ga-NODAGA-c(NGR) (MG1)	−2.33 ± 0.14
[^68^Ga]Ga-NODAGA-c(NGR) (MG2)	−2.29 ± 0.13
[^68^Ga]Ga-NODAGA-YEVGHRC	−4.421
[^68^Ga]Ga-NOTA-G_3_-NGR	−2.25 ± 0.17
[^68^Ga]Ga-NOTA-G_3_-NGR_2_	−2.76 ± 0.20

*cNGR: cyclic NGR; ^68^Ga: Gallium-68; G_3_-NGR: Gly_3_-CNGRC; LogP: partition coefficient; MG1: c[CH_2_-CO-Lys(NODAGA)-Asn-Gly-Arg-Cys]-NH_2_, MG2: c[CH_2_-CO-Lys(NODAGA)-Asn-N(Me)Gly-Arg-Cys]-NH_2_; NODAGA: 1,4,7-triazacyclononane,1-glutaric acid-4,7-acetic acid; NGR: Asparagine-Glycine-Arginine; NOTA: 1,4,7-triazacyclononane-triacetic acid; YEVGHRC: tyrosyl-glutamyl-valyl-glycyl-histidyl-arginyl-cysteine.*

**Table 3 ijms-24-12675-t003:** Physical characteristics of ^68^Ga and ^64^Cu radionuclides.

Isotope	Half-Life (h)	Production	Decay Method (%)	Decay Product	β^+^ Endpoint Energy, keV (%)	Principal γ Energies, keV (Abs. %)	Positron Range in Water (mm)	Delayed Imaging (>3 h)	Application
^68^Ga	1.13	^68^Ge/^68^Ga generator and cyclotron	EC (11.1) ß^+^ (88.9)	^64^Ni ^64^Zn	1899 (87.7) 822 (1.2)	511 (177.8) 1077 (3.2) 1261 (0.1) 1883 (0.1)	1.2	not possible	PET diagnostics For radiopharmaceuticals requiring short circulation times for targeting; e.g., peptides
^64^Cu	12.7	cyclotron	EC (43.9) ß^+^ (17.6) ß^−^ (38.5)	^68^ Zn	653 (17.6)	511 (35.2) 1346 (0.5)	0.2	possible	PET imaging/therapy (theranostic) For radiopharmaceuticals requiring long circulation times, e.g., slowly localising antibodies

*^64^Cu: copper-64; EC: electron capture; ^68^Ga: gallium-68; ^68^Ge: germanium-68; ^64^Ni: nickel-64; PET: positron emission tomography; ^64^Zn: zinc-64; ^68^Zn: zinc-68.*

## Data Availability

The datasets used and/or analysed during the current study are available from the corresponding author upon reasonable request.

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
