# Peer review of "NGR-Based Radiopharmaceuticals for Angiogenesis Imaging: A Preclinical Review"

_ijms, 2023, doi:10.3390/ijms241612675_

Round 1
Reviewer 1 Report
This is a narrative review of preclinical studies reporting the in vitro characteristics, in vivo biodistribution, and tumor-targeting of angiogenesis with NGR-based radiolabeled peptides for SPECT and PET imaging. The authors also briefly compare the performance of NGR-based radiotracers and RGD-based radiotracers, which is another class of angiogenesis-targeting radiotracers. The authors ended by briefly discussing the evidence supporting the use of NGR-based radiotracers in non-oncological conditions.
Specific comments
1. Abstract: “Angiogenesis plays a crucial role in tumor formation….” In the early stage of tumorigenesis, diffusion is typically sufficient for tumor cell nutrition. Angiogenesis becomes vital for tumor growth at a more advanced tumor stage. This highlighted statement may be inaccurate and should be rephrased.
2. Except where it is necessary as when it makes a meaningful contribution to the discussion, I believe it is not necessary to present a detailed description of the methodology of each study. For example, lines 175 to 195 is an entire paragraph devoted to describing the methodology of a single study (Trencsényi et al., 2009).
3. On page 8, the authors describe the conflicting findings from different studies of the comparative tumor-to-background uptake of APN/CD13-targeting Ga-68-labeled radiopharmaceuticals versus RGD-based tracers for tumor angiogenesis imaging. Could these differences be resolved by considering the affinity of the radiotracers used in the different studies? Like the NGR-based radiotracers, the RGD-based radiotracers are also diverse with differences in affinity for their targets.
4. Section 2.1: Differences between Ga-68-labelled radiotracers and Cu-64-labelled radiotracers discussed from lines 372 to 398 distort the flow of this work. This comparison may be better put at the of the discussion in section 2.2.
5. Section 2.1: The section stretches from page 3 through page 15 in one continuous series of paragraph after paragraph. This makes a difficult reading. I suggest that the authors consider structuring this section into subheadings. Some of the subheadings to consider may include tracer uptake in primary tumors, tracer uptake in metastatic tumors, the temporal evolution of radiotracer uptake with tumor growth, comparison in in-vitro and in vivo properties between the various Ga-68-labelled NGR-based radiotracers, comparison in the vitro and in vivo properties between NGR and RDG-based radiotracers, etc.
6. Section 2.2: It may be more appropriate to present a brief discussion of the comparable properties of Cu-64 versus Ga-68 at the beginning of this section (physical properties and imaging characteristics). This may provide readers with background knowledge of the differences that may be seen in studies comparing peptides labeled with either of these radiometals.
7. The title and aim of this review suggest that targeting of APN/CD13 with NGR-based radiopharmaceuticals for imaging will be reviewed. However, studies discussing therapy based on these targets were also discussed. Please harmonize things – either modify the aim and title of the review or restrict the discussion to just imaging.
8. The manuscript as presented is a summary of several published studies on the subject without much of the authors’ thoughts on the findings from the study or the robustness of the methodology. A review article should be a qualitative synthesis of evidence drawing inferences from published studies. This important point needs to be addressed in overhauling this manuscript to make it suitable for publication and beneficial to potential readers.
9. There are several syntax errors all over the manuscript. I advise the authors to seek the help of a native English speaker to help correct these errors.
1. There are several syntax errors all over the manuscript. I advise the authors to seek the help of a native English speaker to help correct these errors.
Reviewer 2 Report
The selective binding affinity of asparagine-glycine-arginine (NGR) motif to aminopeptidase N (APN/CD13), an important angiogenesis biomarker, makes radiolabelled NGR peptides promising PET/SPECT imaging tools for the non-invasive imaging of APN/CD13 overexpressing malignancies.
The manuscript entitled “NGR-based radiopharmaceuticals for angiogenesis imaging: a preclinical review” is a clear and comprehensive overview on the recent preclinical studies with radiolabelled peptides bearing the NGR motif that have been specially developed for angiogenesis imaging.
The topic of this review is very interesting and relevant to the scientific community and provides a good contribution to the knowledge in the field. The abstract is adequately descriptive. The presented details are sufficient to provide a clear understanding of state-of -the-art and are well supported by pertinent literature. Tables show the data suitably and are relatively easy to interpret and understand. The cited references are relevant and appropriate, and mostly recent publications.
Thus, it is with these considerations in mind that I recommend publication of the manuscript in IJMS after minor revisions.
However, there are some comments that I would like to address to the authors
Body of text
Although the manuscript is presented in a well-structured manner, both section 2 and section 3 are too long. Some parts of the text are difficult to read/understand because the text is too dense, contains a lot of information. Increasing the spacing between selected paragraphs (details of each radiolabeled agent) would facilitate the reading/understanding of the manuscript
For a clearer, reading data concerning Log P values of the targeting vectors (page 7, line151-164) should be withdrawn from the text and presented in a table.
Abbreviations
Perhaps, the authors should list-out the abbreviations separately with suitable explanation for easier understanding. Suitable explanation is still missing for some abbreviations (e.g. DAR, line 261)
Tables
The details of each radiolabelled probe are appropriately presented in table format (Table 1&2) However, both tables should be better formatted in order that data could be displayed more clearly. Also, Table 1 and Table 2 should have similar column headings (for example: Investigated object; Investigated phenomenon/initiatives; Radiopharmaceutical; in vitro and in vivo methods; Reference). Moreover, Table 2 should be arranged to follow the sequence of the details of the radioactive agents described in the text.
Why did the authors presented 64Cu-labelled peptide (PET)in table 2 and not in table 1 together with 68Ga-peptides ?
An additional column in both tables related to remarkable observations or outcomes would be very useful and render the table more interesting to readers.
All peptides referred in the manuscript and, in the tables, should be unequivocally identified by the respective three-letter or one-letter code assigned to them for better understanding. Also, for a better identification of the labelled agent its name (structure) should not be separated and should be written in the same line in the table/text.
references
References must be revised to correct the format. According to IJMS guidelines, in the text, reference numbers should be placed in square brackets and placed before the punctuation. Also, in the reference section the reference numbers should be assigned to each citation.
Example: 1. Author 1, A.B.; Author 2, C.D. Title of the article. Abbreviated Journal Name, Year, Volume, page range; DOI:10.1021/c160024a013.
Round 2
Reviewer 1 Report
Thank you for revising your manuscript. I have no further comments.